# DisenGCD: A Meta Multigraph-assisted Disentangled Graph Learning Framework for Cognitive Diagnosis

**Shangshang Yang**[1,2]    **Mingyang Chen**[1,3]    **Ziwen Wang**[1]    **Xiaoshan Yu**[1]

**Panpan Zhang**[1]        **Haiping Ma**[1,3*]

**Xingyi Zhang**[1,3]

[1]Key Laboratory of Intelligent Computing and Signal Processing of Ministry of Education,
Anhui University, Hefei, Anhui 230601, P. R. China
[2] Anhui Province Key Laboratory of Intelligent Computing and Applications,
[3]Department of Information Materials and Intelligent Sensing Laboratory of Anhui Province
`{yangshang0308, wzw12sir, yxsleo, zppan55, xyzhanghust}@gmail.com`
`q22201127@stu.ahu.edu.cn   hpma@ahu.edu.cn`

## Abstract

Existing graph learning-based cognitive diagnosis (CD) methods have made relatively good results, but their student, exercise, and concept representations are learned and exchanged in an implicit unified graph, which makes the interaction-agnostic exercise and concept representations be learned poorly, failing to provide high robustness against noise in students' interactions. Besides, lower-order exercise latent representations obtained in shallow layers are not well explored when learning the student representation. To tackle the issues, this paper suggests a meta multigraph-assisted disentangled graph learning framework for CD (DisenGCD), which learns three types of representations on three disentangled graphs: student-exercise-concept interaction, exercise-concept relation, and concept dependency graphs, respectively. Specifically, the latter two graphs are first disentangled from the interaction graph. Then, the student representation is learned from the interaction graph by a devised meta multigraph learning module; multiple learnable propagation paths in this module enable current student latent representation to access lower-order exercise latent representations, which can lead to more effective nad robust student representations learned; the exercise and concept representations are learned on the relation and dependency graphs by graph attention modules. Finally, a novel diagnostic function is devised to handle three disentangled representations for prediction. Experiments show better performance and robustness of DisenGCD than state-of-the-art CD methods and demonstrate the effectiveness of the disentangled learning framework and meta multigraph module. The source code is available at https://github.com/BIMK/Intelligent-Education/tree/main/DisenGCD.

## 1 Introduction

In the realm of intelligent education [43, 27, 47], cognitive diagnosis (CD) plays a crucial role in estimating students' mastery/proficiency on each knowledge concept [1], which mainly models the exercising process of students by predicting students' responses based on their response records/logs

---

*Corresponding author.

38th Conference on Neural Information Processing Systems (NeurIPS 2024).

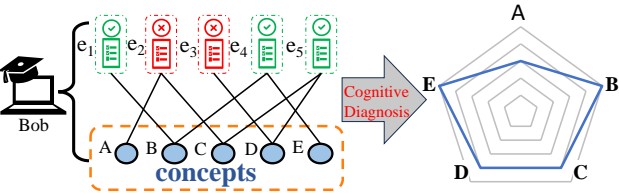

Figure 1: Cognitive diagnosis based on a student's response records and exercise-concept relations.

and the exercise-concept relations. As shown in Figure 1, student Bob completed five exercises $\{e_1, e_2, e_3, e_4, e_5\}$ and got corresponding responses, and his diagnosis result can be obtained through CD based on his records and the plotted relations between exercises and concepts. With students' diagnosis results, many intelligent education tasks can be benefited, such as exercise assembly [42], course recommendation [34, 45], student testing [18], and targeted training [2], and remedial instruction [11].

In recent years, many cognitive diagnosis models (CDMs) have been suggested based on various techniques, generally classified into two types. The first type of CDMs is based on the theory of educational psychology, and the representatives include DINA [4], (IRT) [13], and MIRT [20]. Considering that the simple diagnostic functions of this type of models fail to model complex student interactions, recently, artificial intelligence researchers have tried to employ neural networks (NNs) to enhance and invent novel CDMs. Therefore, the second type of CDMs can be further divided into two sub-types: focusing on inventing novel diagnostic functions and enhancing the input representations of CDMs, respectively. The representatives of the first sub-type of CDMs include the well-known NCD [25], KSCD [15], and so on [44, 16, 12, 36, 38]. For the second sub-type of CDMs, many NN techniques are employed, including hierarchical attention NNs for ECD [48], long short-term memory networks for DIRT [3], graph neural networks (GNNs) for RCD [7], SCD [28], and GraphCDM [21].

Among existing CDMs, GNN-based ones exhibit significantly better performance than others, which is attributed to the rich information propagation and exchange brought by GNNs. These CDMs generally learn the student, exercise, and concept representations implicitly or explicitly on a unified graph [7, 28], which makes three types of representations be exchanged and aggregated fully even if they are distant regarding relations, thus generating better representations and providing better performance. However, learning exercise and concept representations should be student-interaction-agnostic [31], while the above CDMs' learning for exercise and concept representations is easily affected by students' interactions, and the representations will be learned poorly especially when there exists noise in students' interactions. In short, existing GNN-based CDMs fail to provide high robustness against student interaction noise. In addition, these CDMs do not explore the use of lower-order exercise latent representations in shallow layers for learning the student representation due to the intrinsic defect of their GNNs. The learned student representations could be more robust and not sensitive to student interaction data change.

Therefore, we propose a meta multigraph-assisted disentangled graph learning framework for CD (DisenGCD) to learn robust representations against interaction noise. The contributions include

(1) The disentangled graph learning framework DisenGCD learns three types of representations on three disentangled graphs. Specifically, the student representation is learned on the student-exercise-concept interaction graph; the exercise and concept representations are learned on two disentangled graphs: the exercise-concept relation graph and the concept dependency graph. By doing so, the latter two learned representations are interaction-agnostic and robust against student interaction noise.

(2) To make the best of lower-order exercise latent representations for learning the robust student representation, a modified meta multigraph module containing multiple learnable propagation paths is used, where propagation paths enable current student latent representation to access and use lower-order exercise latent representations. The exercise and concept representations are learned through common graph attention networks (GAT) on relation and dependency graphs, respectively. Finally, a novel diagnostic function is devised to handle three learned representations for prediction.

(3) Extensive experiments show the superiority of the proposed DisenGCD to state-of-the-art (SOTA) models regarding performance and robustness, and the effectiveness of the disentangled graph learning framework and the devised meta multigraph module is validated.

## 2 Related Work

**Related Work on Cognitive Diagnosis**. The above has given the classification of CDMs, and we will briefly introduce typical CDMs, especially GNN-based ones. As representatives in educational psychology, IRT [13] (MIRT [20]) utilizes single (multiple) variable(s) to denote student's ability with logistic function for prediction. For NN-based representatives, NCD [25] and KSCD [15] fed student, exercise, and concept vectors to an IRT-like NN as the diagnostic function for prediction; NAS-GCD [44] is similar to NCD, but its diagnostic function is automatically obtained.

For GNN-based CDMs, their focus is on obtaining enhanced representations. For example, RCD [7] learns the student representation along the relation of adjacent exercise nodes, learns the exercise representation along the relation of adjacent student and concept nodes, and learns the concept representation along the relation of adjacent exercise and concept nodes. Since each node's information can be propagated to any node, learning regarding three relations can be seen as learning on an implicit unified graph; Similarly, SCD [28] takes the same aggregation manner along the first two relations of RCD for learning student and exercise representations for contrastive learning to mitigate long-tail problems. Despite the success of GNN-based CDMs, their learning manners of exercise and concept representations are not robust against student interaction noise, because the representations are learned together with student interactions in a unified graph, where the noise will prevent the representations from being learned well.

**Related Work on Meta Graph/Multigraph**. As can be seen, existing GNN-based CDMs intuitively adopt classical GNNs (GAT [24] or GCN[9]) to learn three types of representations. These CDMs update the student representation by only using the exercise latent representation in the previous layer yet not exploring the use of lower-order exercise latent representations.

Compared to traditional GNNs, meta graph-based GNNs can make the target type of nodes access lower-order latent representations of their adjacent nodes. The meta graph is a directed acyclic graph built for a GNN, each node stores each layer's output (latent representations) of the GNN, and each edge between two nodes could be one of multiple types of propagation paths. The representatives include DiffMG [5] and GEMS [8]. The meta multigraph is the same as the meta graph, but its each edge could hold more than one type of propagation path, where the representative is PMMM [10]. To make the best of low-order exercise latent representations for learning effective and robust student representations, this paper adopts the idea of meta multigraph and devises a modified meta multigraph learning module for the updating of the student interaction graph.

**Related Work on Disentangling Graph Representation Learning**. Learning potential representations of disentangling in complex graphs to achieve model robustness and interpretability has been a hot topic in recent years. Researchers have put forward many disentanglement approaches (e.g., DisenGCN [19], DisenHAN [32], DGCF [29], DCCF [35], DcRec [33], etc.) to address this challenge. For example, in DisenHAN [32], the authors utilized disentangled representation learning to account for the influence of each factor in an item. They achieved this by mapping the representation into different spatial dimensions and aggregating item information from various edge types within the graph neural network to extract features from different aspects; DcRec [33] disentangles the network into a user-item domain and a user-user social domain, generating two views through data augmentation and ultimately obtaining a more robust representation via contrastive learning.

Despite many approaches suggested, they were primarily applied to bipartite graphs to learn different representations from different perspectives, for learning more comprehensive representations. While this paper aims to leverage disentanglement learning to mitigate the influence of the interaction noise in the interaction graph, and thus we proposed a meta multigraph-assisted disentangled graph cognitive diagnostic framework to learn three types of representations on three disentangled graphs. By doing so, the influence of the noise on exercise and concept learning can be well alleviated.

## 3 Problem Formulation

For easy understanding, two tables are created to describe all notations utilized in this paper including notations for disentangled graphs and notations for the meta multigraph, summarized in Table 4 and Table 5. Due to the page limit, the two tables are included in **Appendix** A.1.

## 3.1 Disentangled Graph

This paper only employs the student-exercise-concept interaction graph $\mathcal{G}_\mathcal{I}$ for students' representation learning, and disentangles two graphs from $\mathcal{G}_\mathcal{I}$ for the remaining two types of representation learning, which are the exercise-concept relation graph $\mathcal{G}_\mathcal{R}$, and the concept dependency graph $\mathcal{G}_\mathcal{D}$.

**Student-Exercise-Concept Interaction Graph.** With students' response records $\mathcal{R}$, exercise-concept relation matrix, and concept dependency matrix, the interaction graph $\mathcal{G}_\mathcal{I}$ can be represented as $\mathcal{G}_\mathcal{I} = \{\mathcal{V}, \mathcal{E}\}$. The node set $\mathcal{V} = S \cup E \cup C$ is the union of the student set $S$, exercise set $E$, and concept set $C$, while the edge set $\mathcal{E} = \mathcal{R}_{se} \cup \mathcal{R}_{ec} \cup \mathcal{R}_{cc}$ contains three types of relations: $rse_{ij} \in R_{se}$ represents the student $s_i \in S$ answered exercise $e_j \in E$, $rec_{jk} \in R_{ec}$ denotes the exercise $e_j$ contains concept $c_k \in C$, and $rcc_{km} \in R_{cc}$ denotes concept $c_k$ relies on concept $c_m$.

**Disentangled Relation Graph.** To avoid the impact of students' interactions $\mathcal{R}_{se}$ on exercise representation learning, the exercise-concept relation graph $\mathcal{G}_\mathcal{R}$ is disentangled from $\mathcal{G}_\mathcal{I}$, denoted as $\mathcal{G}_\mathcal{R} = \mathcal{G}_\mathcal{I}/\{S, \mathcal{R}_{se}\} = \{E \cup C, \mathcal{R}_{ec} \cup \mathcal{R}_{cc}\}$.

**Disentangled Dependency Graph.** Similarly, a concept dependency graph $\mathcal{G}_\mathcal{D}$ is further disentangled from $\mathcal{G}_\mathcal{I}$ to learn the concept representation without the influence of interactions $\mathcal{R}_{se}$ and exercises' relation $\mathcal{R}_{ec}$, which is represented by $\mathcal{G}_\mathcal{D} = \mathcal{G}_\mathcal{R}/\{E, \mathcal{R}_{ec}\} = \{C, \mathcal{R}_{cc}\}$.

## 3.2 Problem Statement

For the cognitive diagnosis task in an intelligent education online platform, there are usually three sets of items: a set of $N$ students $S = \{s_1, s_2, \ldots, s_N\}$, a set of $M$ exercises $E = \{e_1, e_2, \ldots, e_M\}$, and a set of $K$ knowledge concepts (concept for short) $C = \{c_1, c_2, \ldots, c_K\}$. Besides, there commonly exists two matrices: exercise-concept relation matrix $Q = (Q_{jk} \in \{0, 1\})^{M \times K}$ (Q-matrix) and concept dependency matrix $D = (Q_{km} \in \{0, 1\})^{K \times K}$, to show the relationship of exercises to concepts and concepts to concepts, respectively. $Q_{jk} = 1$ denotes the concept $c_k$ is not included in the exercise $e_j$ and $Q_{jk} = 0$ otherwise; similarly, $D_{km} = 1$ denotes concept $c_k$ relies on concept $c_m$ and $D_{km} = 0$ otherwise. All students' exercising reponse logs are denoted by $\mathcal{R} = \{(s_i, e_j, r_{ij})|s_i \in S, e_j \in E, r_{ij} \in \{0, 1\}\}$, where $r_{ij}$ refers to the response/answer of student $s_i$ on exercise $e_j$. $r_{ij} = 1$ means the answer is correct and $r_{ij} = 0$ otherwise.

The CD based on disentangled graphs is defined as follows: **Given**: students' response logs $\mathcal{R}$ and three disentangled graphs: interaction graph $\mathcal{G}_\mathcal{I}$, relation graph $\mathcal{G}_\mathcal{R}$, and dependency graph $\mathcal{G}_\mathcal{D}$; **Goal**: revealing students' proficiency on concepts by predicting students' responses through NNs.

## 4 Method

For better understanding, Figure 2 presents the overall architecture of the proposed DisenGCD, where three learning modules are used for learning three types of representations on three disentangled graphs and the diagnostic function makes the final prediction based on the learned representations. Specifically, these three learning modules are: (1) a meta multigraph-based student learning module, (2) a GAT-based exercise learning module, and (3) a GAT-based concept learning module. In each learning module, the corresponding graph's embedding is randomly first initialized. Then, the meta multigraph-based student learning module is employed to learn the student representation $\overline{\mathbf{S}_i}$ based on interaction graph $\mathcal{G}_\mathcal{I}$; the GAT-based exercise learning module is used to learn the exercise representation $\overline{\mathbf{E}_j}$ based on relation graph $\mathcal{G}_\mathcal{R}$; while the concept representation $\overline{\mathbf{C}_k}$ is learned by the GAT-based concept learning module on dependency graph $\mathcal{G}_\mathcal{D}$ or the naive embedding if $\mathcal{G}_\mathcal{D}$ is unavailable. Finally, a devised novel diagnostic function receives three learned representations $\overline{\mathbf{S}_i}$, $\overline{\mathbf{E}_j}$, and $\overline{\mathbf{C}_k}$ to get the prediction $\hat{r_{ij}}$ of student $s_i$ on exercise $e_j$.

Note that the first module employs a modified meta multigraph aggregator to learn student representations. Compared to traditional graph learning, the module contains multiple learnable propagation paths, which enable the student latent representation to be learned currently to access and use lower-order exercise latent representations, leading to a more effective and robust student representation.

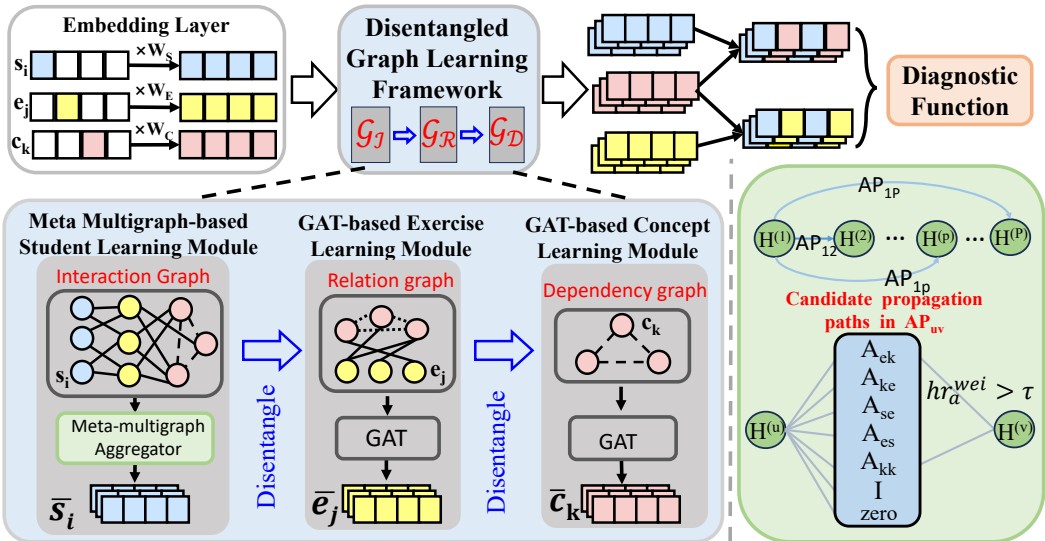

Figure 2: Overview of the proposed DisenGCD: the disentangled graph learning framework is composed of three learning modules: two GAT-based learning modules and a meta multigraph-based learning module, where the latter's details are shown in the green part.

## 4.1 The Meta Multigraph-based Student Learning Module

Based on interaction graph $\mathcal{G}_{\mathcal{I}}$, this module is responsible for: learning the optimal meta multigraph structure in $\mathcal{G}_{\mathcal{I}}$, i.e., optimal multiple propagation paths, and updating student nodes' representations based on the learned meta multigraph structure to get the representation $\overline{\mathbf{S_i}} \in \mathbb{R}^{1 \times d}$.

To start with, all nodes (including students, exercises, and concepts) need to be mapped into a $d$-dimensional hidden space. The student embedding $\mathbf{s}_i \in \mathbb{R}^{1 \times d}$ for student $s_i$, exercise embedding $\mathbf{e}_j^I \in \mathbb{R}^{1 \times d}$ for exercise $e_j$, and concept embedding $\mathbf{c}_k^I \in \mathbb{R}^{1 \times d}$ for concept $c_k$ can be obtained by

$$\mathbf{s}_i = \mathbf{x}_i^S \times W_S^I, \mathbf{e}_j^I = \mathbf{x}_j^E \times W_E^I, \mathbf{c}_k^I = \mathbf{x}_k^C \times W_C^I, W_S^I \in \mathbb{R}^{N \times d}, W_E^I \in \mathbb{R}^{M \times d}, W_C^I \in \mathbb{R}^{K \times d}. \quad (1)$$

$\mathbf{x}_i^S \in \{0,1\}^{1 \times N}$, $\mathbf{x}_j^E \in \{0,1\}^{1 \times M}$, and $\mathbf{x}_k^C \in \{0,1\}^{1 \times K}$ are three one-hot vectors for student $s_i$, exercise $e_j$, concept $c_k$, respecitvely. $W_S^I$, $W_E^I$, and $W_C^I$ are three learnable parameter matrices.

### 4.1.1 The Meta Multigraph Aggregator

To learn more useful representations from the above initial embedding, we construct a meta multigraph $\mathcal{MG} = \{\mathbf{H}, \mathcal{AP}\}$ to specify a set of propagation paths for all nodes' aggregation in graph learning.

As shown in Figure 2, the meta multigraph is actually a directed acyclic graph containing $P$ hyper-nodes: $\mathbf{H} = \{\mathbf{H}^{(1)}, \ldots, \mathbf{H}^{(p)}, \ldots, \mathbf{H}^{(P)}\}$. $\mathbf{H}^{(p)} = \{\mathbf{s}^p \in \mathbb{R}^{N \times d}, \mathbf{e}^p \in \mathbb{R}^{M \times d}, \mathbf{c}^p \in \mathbb{R}^{K \times d}\}$ is a set of latent representations stored in $p$-th hyper-node for all students, exercises, and concepts, whose $i$-th row $\mathbf{s}_i^p \in \mathbb{R}^{1 \times d}$, $j$-th row $\mathbf{e}_j^p \in \mathbb{R}^{1 \times d}$, and $k$-th row $\mathbf{c}_k^p \in \mathbb{R}^{1 \times d}$ are the $p$-th latent representations of student $s_i$, exercise $e_j$, and concept $c_k$, respectively. Especially when $p=1$, $\mathbf{s}_i^p$, $\mathbf{e}_j^p$, and $\mathbf{c}_k^p$ equal initial embedding $\mathbf{s}_i$, $\mathbf{e}_j^I$, and $\mathbf{c}_k^I$.

Each edge $AP_{uv} \in \mathcal{AP}$ between two hyper-nodes $H^{(u)}$ and $H^{(v)}$ contains multiple types ($|HR|$) of propagation paths, which can be denoted as $AP_{uv} = \{(hr_a \in \mathcal{HR}, hr_a^{wei} \in [0,1]) | 1 \le a \le |HR|\}$. Here $hr_a$ is $a$-th type of propagation path in candidate path set $HR$, and $hr_a^{wei}$ represents the weight of $hr_a$. This paper adopts $|HR|=7$ types of propagation paths as the candidate paths for $HR$: 1) the students to exercises path $A_{se}$, 2) the exercises to students path $A_{es}$, 3) the exercises to concepts path $A_{ek}$, 4) the concepts to exercises path $A_{ke}$, 5) the concepts to concepts path $A_{kk}$, 6) the identity path $I$, 7) the zero path $zero$.

Here, path $A_{se}$ means updating the exercise latent representation along the propagation path of student nodes to exercise nodes, and other paths hold similar meanings. As a result, $HR = \{A_{se}, A_{es}, A_{ek}, A_{ke}, A_{kk}, I, zero\}$, and the edge set $\mathcal{AP}$ in $\mathcal{MG}$ can be denoted by $\mathcal{AP} = \{AP_{uv} | 1 \le u < v \le P\}$, containing $|HR| * [P * (P-1)/2]$ propagation paths. With above propagation paths, $\mathbf{H}^{(p)}$ can be updated by

$$\mathbf{H}^{(p)} = \sum \{f(\mathcal{AP}_{up}, \mathbf{H}^{(u)}) | 1 \le u < p\}, \tag{2}$$

where $f(\cdot)$ is the aggregation function of GCN. $f(\mathcal{AP}_{up}, \mathbf{H}^{(u)})$ refers to that obtaining the latent representations $(\mathbf{s}_i^p, \mathbf{e}_j^p, \mathbf{c}_k^p)$ in hyper-node $\mathbf{H}^{(p)}$ based on all latent representations stored in previous hyper-nodes (i.e., $\mathbf{H}^{(1)}$ to $\mathbf{H}^{(p-1)}$). It can be seen that the updating process of $\mathbf{H}^{(p)}$ replies on latent representations in multiple hyper-nodes and multiple propagation paths, which compose **the modified meta multigraph aggregator** together.

For better understanding, here is an example: when $p$=3, $AP_{13} = \{zero\}$ and $AP_{23} = \{A_{es}, I\}$, the updating of three latent representations $\mathbf{s}_i^3$, $\mathbf{e}_j^3$, and $\mathbf{c}_k^3$ is denoted as

$$AP_{13} : \left\{ \mathbf{s}_i^3(13) = 0 * \mathbf{s}_i^1, \mathbf{e}_j^3(13) = 0 * \mathbf{e}_j^1, \mathbf{c}_k^3(13) = 0 * \mathbf{c}_k^1 \right.$$

$$AP_{23} : \begin{cases} \mathbf{s}_i^3(23) = Up(\mathbf{s}_i^2, \sum_{j \in N_{s_i}} Mess(\mathbf{s}_i^2, e_j^2)) \\ \mathbf{e}_j^3(23) = \mathbf{e}_j^2, \mathbf{c}_k^3(23) = \mathbf{c}_k^2 \end{cases} \tag{3}$$

$$\mathbf{s}_i^3 = \mathbf{s}_i^3(13) + \mathbf{s}_i^3(23), \ \mathbf{e}_j^3 = \mathbf{e}_j^3(13) + \mathbf{e}_j^3(23), \ \mathbf{c}_k^3 = \mathbf{c}_k^3(13) + \mathbf{c}_k^3(23)$$

The first equation is to update three latent representations according to $AP_{13}$, while the second is to get the updating based on $AP_{23}$. The final latent representations are obtained by summing up both updated ones. Here $Up(\cdot)$ and $Mess(\cdot)$ are common update and message-passing functions.

### 4.1.2 Routing Strategy

To find suitable propagation paths, threshold $\tau^{(u,v)}$ is created for each pair of hyper-nodes $(u, v)$:

$$\tau^{(u,v)} = \lambda \cdot \max(Softmax(AP_{uv})) + (1 - \lambda) \cdot \min(Softmax(AP_{uv})), \ \lambda \text{ is predefined.} \tag{4}$$

$Softmax(AP_{uv}))$ normalizes weights of each type of propagation path regarding their $hr_a^{wei}$ values. By doing so, the propagation paths of each pair of hyper-nodes will remain if their $hr_a^{wei}$ values are greater than the corresponding threshold. Thus the updating process of $\mathbf{H}^{(p)}$ can be rewritten as

$$\mathbf{H}^{(p)} = \sum \{f(A\hat{P}_{up}, \boldsymbol{H}^{(u)}) \| 1 \le u < p\}$$
$$A\hat{P}_{up} = \{(hr_a, hr_a^{wei}) | hr_a^{wei} \ge \tau^{(u,p)}, \forall hr_a \in AP_{up}\} \tag{5}$$

Finally, the learned student representation $\mathbf{s}^P$ in $\mathbf{H}^P$ are used for the diagnosis, i.e, $\mathbf{s}_i^P$ is used as $\overline{\mathbf{S}}_i$.

## 4.2 The GAT-based Exercise Learning Module and GAT-based Concept Learning Module

**GAT-based Exercise Learning Module**. This module is responsible for learning the exercise representation $\overline{\mathbf{E_j}} \in \mathbb{R}^{1 \times d}$ on the relation graph $\mathcal{G}_\mathcal{R}$ via a $L$-layer GAT network [24]. Firstly, the embedding of exercises and concepts in $\mathcal{G}_\mathcal{R}$ is obtained in the manner same as Eq.(1) through two learnable matrices $W_E^R \in \mathbb{R}^{M \times d}$ and $W_C^R \in \mathbb{R}^{K \times d}$, i.e, $\mathbf{e}_j^C = \mathbf{x}_j^E \times W_E^R, \mathbf{c}_k^R = \mathbf{x}_k^C \times W_C^R$.

Afterward, the GAT neural network is applied to aggregate neighbor information to learn the exercise representation. The aggregation process of $l$-th layer $(1 \le l \le L)$ can be represented as

$$\mathbf{e}_j^{R(l)} = \sum_{k \in N_{e_j}} \alpha_{j(k)}^{R(l)} \mathbf{c}_k^{R(l-1)} + \mathbf{e}_j^{R(l-1)},$$
$$\mathbf{c}_k^{R(l)} = \sum_{j \in N_{c_k}^{ec}} \alpha_{k(j)}^{R(l)} \mathbf{e}_j^{R(l-1)} + \sum_{m \in N_{c_k}^{cc}} \alpha_{\hat{k}(m)}^{R(l)} \mathbf{c}_m^{R(l-1)} + \mathbf{c}_k^{R(l-1)} \tag{6}$$

The first equation is to aggregate the information of exercise $e_j$'s neighbors $N_{e_j}$ to get its $l$-th layer's latent representation $\mathbf{e}_j^{R(l)} \in \mathbb{R}^{1 \times d}$, while the second is to update the $l$-th layer's concept latent representation $\mathbf{c}_k^{R(l)} \in \mathbb{R}^{1 \times d}$ from its exercise neighbors $N_{c_k}^{ec}$ and concept neighbors $N_{c_k}^{cc}$. $\alpha_j^{R(l)}$,

$\alpha_k^{R(l)}$, and $\alpha_{\hat{k}}^{R(l)}$ are the $l$-th layer's attention matrices for exercise $e_j$'s concept neighbors, concept $c_k$'s exercise neighbors, and $c_k$'s concept neighbors. They can be obtained in the same manner, and the $k$-th row of $\alpha_j^{R(l)}$ can be obtained by

$$\alpha_{j(k)}^{R(l)} = \text{Softmax}(F_{ec}([\mathbf{e}_j^{R(l-1)}, \mathbf{c}_k^{R(l-1)}])), \forall k \in N_{e_j}. \tag{7}$$

$[\cdot]$ is the concatenation and $F_{ec}(\cdot)$ is a fully connected (FC) layer mapping $2 * d$ vectors to scalars.

The latent representation $\mathbf{e}_j^{R(0)}$ and $\mathbf{c}_k^{R(0)}$ refer to $\mathbf{e}_j^R$ and $\mathbf{c}_k^R$, and the $L$-th layer output $\mathbf{e}_j^{R(L)}$ is used as $\overline{\mathbf{E}_j}$. By disentangling the interaction data, the learned exercise representation $\overline{\mathbf{E}_j}$ will be more robust against interaction noise.

**GAT-based Concept Learning Module**. Similarly, this module obtains the concept representation $\overline{\mathbf{C}_k} \in \mathbb{R}^{1 \times d}$ by applying a $L$-layer GAT network to the dependency graph $\mathcal{G}_\mathcal{D}$. After obtaining the initial embedding $\mathbf{c}_k^D = \mathbf{x}_k^C \times W_C^D$ by a learnable matrix $W_C^D \in \mathbb{R}^{K \times d}$, this module updates the $l$-th layer's latent representation $\mathbf{c}_k^{D(l)}$ by $\mathbf{c}_k^{D(l)} = \sum_{m \in N_{c_k}^{cc}} \alpha_{\hat{k}(m)}^{D(l)} \mathbf{c}_m^{D(l-1)} + \mathbf{c}_k^{D(l-1)}$. $\alpha_{\hat{k}}^{D(l)}$ denotes the attention matrix, which can be computed as same as Eq.(7).

Here $\mathbf{c}_k^{D(0)}$ is equal to $\mathbf{c}_k^D$, and the $L$-th layer output $\mathbf{c}_k^{D(L)}$ is used as $\overline{\mathbf{C}_k}$. If graph $\mathcal{G}_\mathcal{D}$ is unavailable, this module will directly take the initial embedding $\mathbf{c}_k^D$ as $\overline{\mathbf{C}_k}$. By further disentangling, the learned representation $\overline{\mathbf{C}_k}$ may be robust against noise in student interactions to some extent.

## 4.3 The Diagnosis Module

To effectively handle the obtained three types of representations, a novel diagnostic function is proposed to predict the response $\hat{r_{ij}}$ of student $s_i$ got on exercise $e_j$ as follows:

$$\mathbf{h}_{simi} = \sigma(F_{simi}(\mathbf{h}_{si} \cdot \mathbf{h}_{ej})), \mathbf{h}_{si} = F_{si}(\overline{\mathbf{S}_i} + \overline{\mathbf{C}_k}), \mathbf{h}_{ej} = F_{ej}(\overline{\mathbf{E}_j} + \overline{\mathbf{C}_k})$$
$$\hat{r_{ij}} = (\sum Q_k \cdot \mathbf{h}_{simi}) / \sum Q_k \tag{8}$$

where $F_{si}(\cdot)$, $F_{ej}(\cdot)$, and $F_{simi}(\cdot)$ are three FC layers mapping a $d$-dimensional vector to another one, and $Q_k$ is a binary vector in the $k$-th row of Q-matrix.

Here $\mathbf{h}_{si} \in \mathbb{R}^{1 \times d}$ can be seen as the student's mastery of each knowledge concept; while the obtaining of $\mathbf{h}_{ej} \in \mathbb{R}^{1 \times d}$ aims to get the exercise difficulty of each concept; $\mathbf{h}_{simi}$ is to measure the similarity between $\mathbf{h}_{si}$ and $\mathbf{h}_{ej}$ via a dot-product followed by an FC layer and a Sigmoid function $\sigma(\cdot)$ [41], where a higher similarity value in each bit represents a higher mastery on each concept, further indicating a higher probability of answering the related exercises; the last equation follows the idea of NCD to compute the overall mastery averaged over all concepts contained in exercise $e_j$.

We can see that the proposed diagnostic function has as high interpretability as NCD, IRT, and MIRT.

**Model Optimization.** With the above modules, the proposed DisenGCD are trained by solving the following bilevel optimization problem through Adam [39]:

$$\min_{\boldsymbol{\alpha}} \mathcal{L}_{val}(D_{val}|\boldsymbol{\omega}^*(\boldsymbol{\alpha}), \boldsymbol{\alpha}), \text{ s.t. } \boldsymbol{\omega}^*(\boldsymbol{\alpha}) = \text{argmin}_{\boldsymbol{\omega}} \mathcal{L}_{train}(D_{train}|\boldsymbol{\omega}, \boldsymbol{\alpha}), \tag{9}$$

where $\mathcal{L}_{val}(\cdot)$ and $\mathcal{L}_{train}(\cdot)$ denote the loss on validation dataset $D_{val}$ and training dataset $D_{train}$. $\boldsymbol{\omega}$ denotes all model parameters, and $\boldsymbol{\alpha}$ denotes the weights of learnable propagation paths $\mathcal{AP}$ in meta multigraph $MG$. The cross-entropy loss [40] is used for $\mathcal{L}_{val}$ and $\mathcal{L}_{train}$.

## 5 Experiments

This section answers the following questions: **RQ1**: How about the performance of DisenGCD compared to SOTA CDMs? **RQ2**: How about the DisenGCD's robustness against noise and the disentangled learning framework's effectiveness in DisenGCD? **RQ3**: How about the effectiveness of the devised meta multigraph learning module? **RQ4**: How does the learned meta multigraph on target datasets look like, and how about their generalization on other datasets? In addition, **more experiments to validate the proposed DisenGCD's effectiveness are in the Appendix.**

Table 1: Statistics of three datasets: ASSISTments, Math, and SLP.

| Datasets | Students | Exercises | Concepts | Logs | Avg logs per student |
|---|---|---|---|---|---|
| # ASSISTments | 4,163 | 17,746 | 123 | 278,868 | 67 |
| # Math / SLP | 1,967 / 1,499 | 1,686 / 907 | 61 / 33 | 118,348 / 57,244 | 60 / 38 |

Table 2: Performance comparison between DisenGCD and five CDM in terms of AUC, ACC, and RMSE values, obtained on ASSISTments and Math. Four dataset-splitting ratios were adopted, and the best result of each column on one dataset was highlighted.

| Datatset | Ratio | 40%/10%/50% | | | 50%/10%/40% | | | 60%/10%/30% | | | 70%/10%/20% | | |
|---|---|---|---|---|---|---|---|---|---|---|---|---|---|
| | Method | ACC↑ | RMSE↓ | AUC↑ | ACC↑ | RMSE↓ | AUC↑ | ACC↑ | RMSE↓ | AUC↑ | ACC↑ | RMSE↓ | AUC↑ |
| ASSISTments | DINA | 0.6388 | 0.4931 | 0.6874 | 0.6503 | 0.4862 | 0.4978 | 0.6573 | 0.4820 | 0.7071 | 0.6623 | 0.4787 | 0.7126 |
| | MIRT | 0.6954 | 0.4740 | 0.7254 | 0.7015 | 0.4689 | 0.7358 | 0.7096 | 0.4624 | 0.7469 | 0.7110 | 0.4617 | 0.7514 |
| | NCD | 0.7070 | 0.4443 | 0.7374 | 0.7142 | 0.4370 | 0.7423 | 0.7237 | 0.4365 | 0.7552 | 0.7285 | 0.4298 | 0.7603 |
| | ECD | 0.7154 | 0.4373 | 0.7362 | 0.7130 | 0.4373 | 0.7432 | 0.7274 | 0.4329 | 0.7543 | 0.7297 | 0.4296 | 0.7599 |
| | RCD | 0.7232 | 0.4311 | 0.7546 | 0.7253 | 0.4285 | 0.7605 | 0.7291 | 0.4262 | 0.7663 | 0.7296 | 0.4245 | 0.7687 |
| | DisenGCD | **0.7276** | **0.4255** | **0.7635** | **0.7287** | **0.4238** | **0.7677** | **0.7335** | **0.4219** | **0.7723** | **0.7334** | **0.4209** | **0.7746** |
| Math | DINA | 0.6691 | 0.4715 | 0.7117 | 0.6745 | 0.4674 | 0.7199 | 0.6813 | 0.4633 | 0.7222 | 0.6812 | 0.4635 | 0.7231 |
| | MIRT | 0.7229 | 0.4335 | 0.7427 | 0.7227 | 0.4299 | 0.7497 | 0.7279 | 0.4291 | 0.7479 | 0.7340 | 0.4256 | 0.7542 |
| | NCD | 0.7394 | 0.4157 | 0.7604 | 0.7424 | 0.4119 | 0.7660 | 0.7418 | 0.4109 | 0.7706 | 0.7447 | 0.4084 | 0.7756 |
| | ECD | 0.7335 | 0.4154 | 0.7615 | 0.7424 | 0.413 | 0.7657 | 0.7434 | 0.4114 | 0.7693 | 0.7484 | 0.4087 | 0.7761 |
| | RCD | 0.7446 | 0.4100 | 0.7724 | 0.7489 | 0.4074 | 0.7751 | 0.7501 | 0.4078 | 0.7806 | 0.7534 | 0.4034 | 0.7866 |
| | DisenGCD | **0.7479** | **0.4076** | **0.7802** | **0.7513** | **0.4052** | **0.7832** | **0.7527** | **0.4039** | **0.7867** | **0.7582** | **0.4004** | **0.7932** |

## 5.1 Experimental Settings

**Datasets.** To verify the DisenGCD's effectiveness, we conducted experiments on two public datasets ASSISTments [6] and SLP [14], and one private dataset Math, whose statistics are in Table 1. Note that ASSISTments and Math [16] were used for most experiments to answer **RQ1** to **RQ3**, and SLP was used to answer **RQ4**. **More details of these datasets are presented in Appendix A.2**.

**Comparison CDMs and Metrics.** To verify the effectiveness of DisenGCD, five SOTA CDMs were compared, including traditional CDMs DINA and MIRT, NN-based CDMs NCD and ECD, and GNN-based RCD. Here SCD is not compared due to the failed run of its provided source code Three metrics were adopted to measure the performance of all CDMs [37], including *area under the cure* (AUC) [23], *accuracy* (ACC) [22], and *root mean square error* (RMSE) [46].

**Parameter Settings.** For the DisenGCD model, its dimension $d$ was set to the number of concepts, $P$ in the meta multigraph, the number of layers $L$ in GAT, and $\lambda$ were set to 5, 2, and 0.8. For its training, the learning rate and batch size were 1e-4 and 256. For comprehensive comparisons [17], four splitting ratios (40%/10%/50%, 50%/10%/40%, 60%/10%/30%, and 70%/10%/20%) were adopted to get training, validation, and testing datasets. All compared CDMs followed the settings in their original papers, and all experiments were executed on an NVIDIA RTX4090 GPU.

## 5.2 Overall Performance Comparison (RQ1)

To address **RQ1**, the DisenGCD was compared with DINA, MIRT, NCD, ECD, and RCD on ASSISTments and Math datasets. Table 2 summarizes their performance in terms of AUC, ACC, and RMSE obtained under four dataset-splitting settings, where the best result of each column on one dataset was highlighted in bold. Besides, Table 6 in the **Appendix** compares DisenGCD with three SOTA CDMs, including SCD [28], KSCD [15], and KaNCD [26].

We can observe from the results in both tables that, DisenGCD holds better performance than all compared CDMs. Besides, we can also obtain two observations from Table 2: Firstly, GNN-based CDMs (RCD and DisenGCD) hold significantly better than NN-based CDMs (NCD and ECD), indicating the importance of learning representations through graphs, and DisenGCD outperforming RCD validates the superiority of DisenGCD's graph learning manner. Secondly, as the ratio changes,

Table 3: Performance of DisenGCD, RCD, and its four variants on ASSISTments dataset.

| Metric | RCD | DisenGCD(I) | DisenGCD (Is+Rec) | DisenGCD (Ise+Rc) | DisenGCD (Isc+Re) | DisenGCD |
|---|---|---|---|---|---|---|
| ACC↑ | 0.7291 | 0.7331 | 0.7321 | 0.7301 | 0.7333 | **0.7335** |
| RMSE↓ | 0.4262 | 0.4235 | 0.4259 | 0.4235 | 0.4231 | **0.4219** |
| AUC↑ | 0.7663 | 0.7678 | 0.7701 | 0.7678 | 0.7685 | **0.7723** |

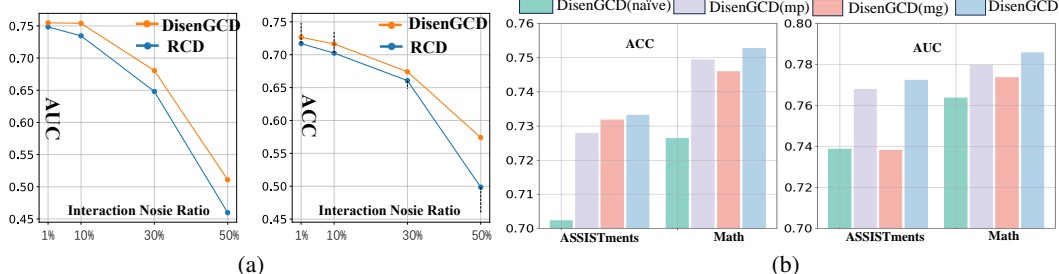

(a)  (b)

Figure 3: (a): Performance of RCD and DisenGCD under different noises. (b): Effectiveness of the meta multigraph learning module.

the performance change of DisenGCD and RCD is much smaller than NCD and ECD, which signifies DisenGCD and RCD are more robust to different-sparsity data. To validate its sparsity superiority, **more experiments** are summarized in **Appendix B.1**.

### 5.3 Effectiveness of Disentangled Learning Framework of DisenGCD (RQ2)

To investigate DisenGCD's robustness against interaction noise, we conducted robust experiments on the ASSISTments dataset under the ratio of 60%/10%/30%, where a certain amount of noise interactions were added to each student in the training and validation datasets. Figure 3a presents the ACC and AUC of DisenGCD and RCD under noise data of different percentages. As the noise data increases, the performance leading of DisenGCD over RCD becomes more significant, which reaches maximal when noise data of 50% was added. That demonstrates DisenGCD is more robust to student noise interactions than RCD, attributed to the disentangled learning framework of DisenGCD. Similar observations can be drawn from **more experiments on other two datasets in Appendix B.2**.

To further analyze the framework effectiveness, four variants of DisenGCD were created: *DisenGCD(I)* refers to learning three representations only on the interaction graph $\mathcal{G}_\mathcal{I}$; *DisenGCD(Is+Rec)* refers to learning student representation on $\mathcal{G}_\mathcal{I}$, but learning exercise and concept ones on the relation graph $\mathcal{G}_\mathcal{R}$; *DisenGCD(Ise+Rc)* refers to learning student and exercise representations on $\mathcal{G}_\mathcal{I}$ but learning concept one on $\mathcal{G}_\mathcal{R}$; while *DisenGCD(Isc+Re)* learns student and concept representations on $\mathcal{G}_\mathcal{I}$ but learns exercise one on $\mathcal{G}_\mathcal{R}$. Table 3 compares the performance of RCD, DisenGCD, and its four variants on ASSISTments. As can be seen, the comparison between *DisenGCD(I)* and other variants indicates learning three representations in two disentangled graphs is more effective than in one unified graph, especially with textitDisenGCD(Is+Rec); the comparison between *DisenGCD(Is+Rec)* and DisenGCD indicates learning three representations in three disentangled graphs is more effective, further validating the above conclusion. Finally, we can conclude that the proposed disentangled learning framework is effective in enhancing DisenGCD's performance and robustness.

### 5.4 Effectiveness of Meta Multigraph Learning Module in DisenGCD (RQ3)

In Table 3, both *DisenGCD(I)* and RCD learn three types of representations in one unified graph, but *DisenGCD(I)* utilizes the meta multigraph aggregator. Therefore, the comparison between *DisenGCD(I)* and RCD proves the devised meta multigraph learning module is effective in improving the model performance to some extent.

To further validate this, we created three variants of DisenGCD: *DisenGCD(naive)* refers to the meta multigraph module replaced by the naive embedding (i.e., $\overline{\mathbf{S_i}}$ equal to $\mathbf{s}_i$ in Eq.(1)); *Dis-*

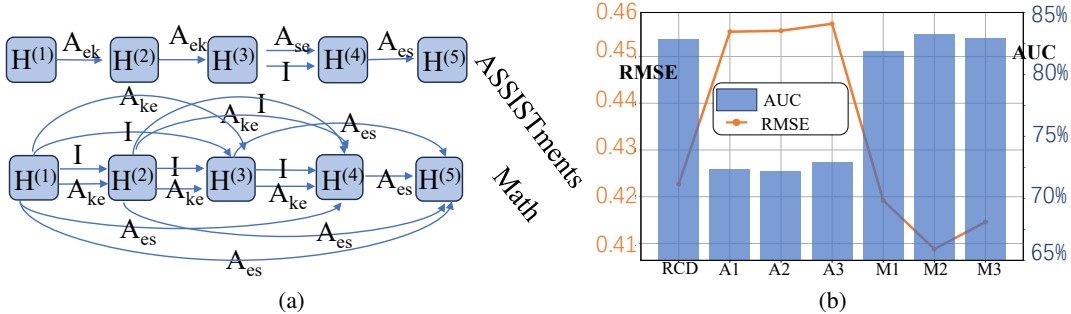

Figure 4: (a): Visualization of learned meta multigraph on two datasets. (b): Generalization validation of six learned meta multigraphs.

*enGCD(mp)* refers to predefining the meta multigraph module's propagation paths according to HAN [30]; *DisenGCD(mg)* refers to this module's meta multigraph replaced by meta graph. Figure 3b presents the AUC and ACC values of DisenGDCN and its variants on two datasets under the ratio of 60%/10%/30%. As can be seen, the devised meta multigraph learning module enables DisenGDCN to hold significantly better performance than the naive embedding-based variant. The comparison results between DisenGCD and *DisenGCD(mg)* as well as *DisenGCD(mp)* validates the effectiveness of using meta multigraph and the automatically learned propagation paths (i.e., learned meta multigraph). Thus, the effectiveness of the devised meta multigraph module can be validated.

### 5.5 Visualization and Effectiveness of Learned Meta Multigraph (RQ4)

The comparison between *DisenGCD(mp)* and DisenGCD has revealed the effectiveness of the learned meta multigraph on the target dataset. Therefore, an intuitive doubt naturally emerged: Is the learned meta multigraph still effective on other datasets? Before solving this, Figure 4a gives the structure visualization of two learned meta multigraphs. We can see two learned meta multigraphs hold two distinct structures. That may be because two datasets are a bit different regarding the relations between exercises and concepts, where exercises in Math contain only one concept while exercises in ASSISTments may contain more than one concept. Thus, another doubt naturally emerged: Is the learned meta multigraph on the target dataset ineffective on a different type of dataset?

To solve the doubts, we applied six learned meta multigraphs (A1, A2, A3, M1, M2, and M3) to the SLP. A1-A3 and M1-M3 were learned by DisenGCD three times on ASSISTments and Math. SLP dataset is similar to Math, whose exercises only contain one concept. Figure 4b summarizes the results of RCD and six DisenGCD variants that utilize six given meta multigraphs. As can be seen, DisenGCD's performance under M1-M3 is promising and better than RCD, while DisenGCD under A1-A3 performed poorly. The observation can answer the above doubts to some extent: meta multigraphs learned by DisenGCD may be effective when the datasets to be applied are similar to target datasets.

In addition to the above four experiments, **more experiments** were executed to validate the devised diagnostic functions, analyze the DisenGCD's parameter sensitivity, analyze its execution efficiency, and investigate the effectiveness of the employed GAT modules in **Appendix B.3**, **Appendix B.4**, **Appendix B.5**, and **Appendix B.6**.

## 6  Conclusion

This paper proposed a meta multigraph-assisted disentangled graph learning framework for CD, called DisenGCD. The proposed DisenGCD learned student, exercise, and concept representations on three disentangled graphs, respectively. It devised a meta multigraph module to learn student representation and employed two common GAT modules to learn exercise and concept representations. Compared to SOTA CDMs on three datasets, the proposed DisenGCD exhibited highly better performance and showed high robustness against interaction noise.

## Acknowledgements

This work was supported in part by the National Key R&D Program of China (No.2018AAA0100100), in part by the National Natural Science Foundation of China (No.62302010, No.62303013, No.62107001, No.62006053, No.61876162, No.62136008, No.62276001, No.U21A20512), in part by China Postdoctoral Science Foundation (No.2023M740015), in part by the Postdoctoral Fellowship Program (Grade B) of China Postdoctoral Science Foundation (No.GZB20240002), and in part by the Anhui Province Key Laboratory of Intelligent Computing and Applications (No. AFZNJS2024KF01).

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

# Appendix A   More details about Proposed Method and Datasets

## A.1   Notations Summary

For convenient reading and understanding, we have summarized all the notations discussed in the paper in the following two tables. Table 4 summarizes all the notations in the three disentangled graphs, while Table 5 describes the notations of the meta multigraph.

Table 4: Notation of disentangled graphs.

| Notation | Description |
| --- | --- |
| $\mathcal{G}_{\mathcal{I}}$ | the student-exercise-concept interaction graph |
| $\mathbf{s}_i$ | initial embedding of student in $\mathcal{G}_{\mathcal{I}}$ |
| $\mathbf{e}_j^I$ | initial embedding of exercise in $\mathcal{G}_{\mathcal{I}}$ |
| $\mathbf{c}_k^I$ | initial embedding of concept in $\mathcal{G}_{\mathcal{I}}$ |
| $\overline{\mathbf{S}}_i$ | the final representation of the updated student node |
| $\mathcal{G}_{\mathcal{R}}$ | the exercise-concept relation Graph |
| $\mathbf{e}_j^R$ | initial embedding of exercise in $\mathcal{G}_{\mathcal{R}}$ |
| $\mathbf{c}_k^R$ | initial embedding of concept in $\mathcal{G}_{\mathcal{R}}$ |
| $\overline{\mathbf{E}}_j$ | the final representation of the updated exercise node |
| $\mathcal{G}_{\mathcal{D}}$ | the concept dependency graph |
| $\mathbf{c}_k^D$ | initial embedding of concept in $\mathcal{G}_{\mathcal{D}}$ |
| $\overline{\mathbf{C}}_k$ | the final representation of the updated concept node |

Table 5: Notation of meta multigraph.

| Notation | Description |
| --- | --- |
| $\mathcal{MG}$ | meta multigraph |
| $\mathbf{H}^{(\mathbf{p})}$ | the p-th hyper node in meta multigraph |
| $\mathcal{AP}$ | edges between hyper nodes(propagation paths) |
| $AP_{uv}$ | edges between $\mathbf{H}^{(\mathbf{u})}$ and $\mathbf{H}^{(\mathbf{v})}$ |
| $HR$ | the set of propagation path types |
| $A_{se}$ | the propagation path from student to exercise |
| $A_{es}$ | the propagation path from exercise to student |
| $A_{ke}$ | the propagation path from concept to exercise |
| $A_{ek}$ | the propagation path from exercise to concept |
| $A_{k\hat{k}}$ | the propagation path from concept to concept |
| $I$ | information is not updated to propagate to the next hyper-node |
| $zero$ | no information propagation between two hyper nodes |
| $hr_a^{wei}$ | the weight of each type propagation path in $AP_{uv}$ |
| $\tau^{(u,v)}$ | the threshold of candidate propagation paths in $AP_{uv}$ |

Table 6: Performance comparison of recent CDMs (SCD, KaNCD, KSCD) and DisenGCD on the Math dataset.

| Method/Metric | SCD | KaNCD | KSCD | DisenGCD |
|---|---|---|---|---|
| ACC | 0.7546 | 0.7519 | 0.7549 | **0.7582** |
| RMSE | 0.4025 | 0.4050 | 0.4040 | **0.4004** |
| AUC | 0.7882 | 0.7854 | 0.7890 | **0.7932** |

## A.2 Statistics of Datasets

We evaluated our method on three real-world datasets: ASSISTments, Math and SLP, which both provide student-exercise interaction records and the exercise-knowledge concept relational matrix.

- **ASSISTments** is a public dataset collected by the assistant online tutoring systems in the 2009-2010 acadaemic year.
- **Math** is a private dataset collected by a well-known online learning platform that contains math practice records and test records for elementary and middle school students.
- **SLP** is another publicdata set that collects data on learners' performance in eight different subjects over three years of study, including maths, English, physics, chemistry, biology, history and geography.

Notably, in both the Math dataset and the SLP dataset, they provide relationships between concepts. For these three datasets, we filtered students with fewer than 15 answer records to ensure there was enough data for the learning of the model. We compared our model with five previous diagnostic models, including two classic models based on educational psychology, DINA [4] and MIRT [20], two neural network-based models, NCD [25] and ECD [48], and a graph-based diagnostic model, RCD [7].

- **DINA** [4] is a classical CDM, which uses a binary variable to characterize whether and whether a student has mastered a specific concept.
- **MIRT** [20] is an extension of the irt model, which uses multidimensional vectors to characterize students' abilities and the difficulty of exercises, and uses linear functions to model the interactions.
- **NCD** [25] is one of the recent CDMs based on deep learning, which uses neural networks to model higher-order student-exercise complex interactions.
- **ECD** [48] incorporates the impact of the educational environment in students' historical answer records into the diagnostic model to achieve diagnostic enhancement. Here due to the lack of educational background, we use random initialization vectors to characterize cognitive states.
- **RCD** [7] is one of the most advanced models, which introduces the relationship between concepts into cognitive diagnosis and models the relationship through a graph structure.

## Appendix B  Additional Experiments

### B.1  Experiments on Dataset with Different Levels of Sparsity

As shown in Section 5.2, the proposed DisenGCD shows better performance than RCD even when the splitting ratios change, which indicates the better robustness of DisenGCD to different-sparsity data.

To further verify whether the proposed DisenGCD is robust against sparse data (i.e., missing data or sparse interaction patterns), we conducted corresponding experiments on ASSISTments and Math datasets under the splitting setting of 60%/ 10% / 30%. In the experiments, for each dataset, we randomly deleted 5%, 10%, and 20% of the students' answer records (i.e., the interaction data) in the training sets, respectively, thus providing three variant training sets; the proposed DisenGCD

Table 7: Performance comparison between DisenGCD, NCD, and RCD under sparse interaction data.

| Datasets | Models | 5% sparsity | | | 10% sparsity | | | 20% sparsity | | |
|---|---|---|---|---|---|---|---|---|---|---|
| | | ACC | RMSE | AUC | ACC | RMSE | AUC | ACC | RMSE | AUC |
| ASSISTments | NCD | 0.7216 | 0.4337 | 0.7487 | 0.7170 | 0.4365 | 0.7455 | 0.7146 | 0.4471 | 0.7398 |
| | RCD | 0.7264 | 0.4273 | 0.7636 | 0.7245 | 0.4295 | 0.7610 | 0.7241 | 0.4289 | 0.7569 |
| | DisenGCD | 0.7282 | 0.4248 | 0.7676 | 0.7278 | 0.4254 | 0.7660 | 0.7273 | 0.4256 | 0.7631 |
| Math | NCD | 0.7399 | 0.4122 | 0.7674 | 0.7383 | 0.4148 | 0.7651 | 0.7352 | 0.4162 | 0.7629 |
| | RCD | 0.7496 | 0.4071 | 0.7790 | 0.7481 | 0.4104 | 0.7781 | 0.7476 | 0.4075 | 0.7771 |
| | DisenGCD | 0.7526 | 0.4030 | 0.7891 | 0.7503 | 0.4049 | 0.7832 | 0.7485 | 0.4058 | 0.7812 |

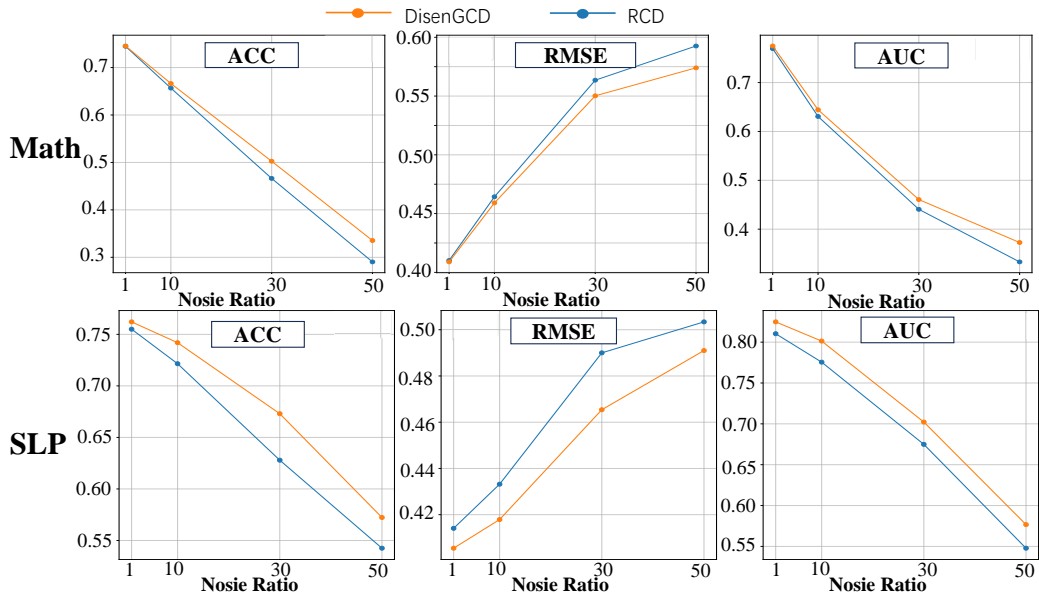

Figure 5: Futher Robustness Validation of DisenGCD.

was compared with NCD and RCD, and their overall performance comparisons were summarized in Table B.1.

As can be seen, under which deleting ratio, the proposed DisenGCD exhibits the best performance, and its performance does not drop very significantly when the deleting ratio increases. Even when deleting 20% of students' interaction data, DisenGCD still achieves an AUC value of 0.7812, which is greatly higher than NCD and RCD. It demonstrates that the proposed DisenGCD is effective on sparse interaction data, which is mainly attributed to the devised meta-multigraph learning module. That module enables the DisenGCD to access and use lower-order exercise latent representations, thus providing more accurate and robust students' representations, especially when partial data is lacking.

In summary, the proposed DisenGCD is effective and more robust against sparse interaction data, which benefits from the devised meta-multigraph learning module, and thus the effectiveness of the meta-multigtaph learning module can be indirectly demonstrated.

## B.2 More Experiments for Robustness Validation

In Section 5.3, we have validated the robustness of the proposed DisenGCD against interaction noise on the ASSISTments dataset. To further show its robustness superiority, we executed the same experiments on other two datasets, i.e., Math and SLP, under the same settings, where a certain ratio of noise interactions were added to each student in the training and validation datasets. Figure 5

Table 8: Validation of the devised diagnostic function on Math dataset.

| Splitting Ratios | 70%/10%/20% | | | 60%/10%/30% | | | 50%/10%/40% | | | 40%/10%/50% | | |
|---|---|---|---|---|---|---|---|---|---|---|---|---|
| Diagnostic Functions | ACC | RMSE | AUC | ACC | RMSE | AUC | ACC | RMSE | AUC | ACC | RMSE | AUC |
| MIRT | 0.7340 | 0.4256 | 0.7542 | 0.7279 | 0.4291 | 0.7479 | 0.7227 | 0.4299 | 0.7497 | 0.7229 | 0.4335 | 0.7427 |
| NCD | 0.7447 | 0.4084 | 0.7756 | 0.7418 | 0.4109 | 0.7706 | 0.7424 | 0.4119 | 0.7660 | 0.7394 | 0.4157 | 0.7604 |
| RCD | 0.7534 | 0.4034 | 0.7866 | 0.7501 | 0.4078 | 0.7806 | 0.7489 | 0.4074 | 0.7751 | 0.7446 | 0.4100 | 0.7724 |
| DisenGCD-MIRT | 0.7396 | 0.4095 | 0.7713 | 0.7385 | 0.4103 | 0.7705 | 0.7378 | 0.4118 | 0.7689 | 0.7367 | 0.4138 | 0.7656 |
| DisenGCD-NCD | 0.7501 | 0.4057 | 0.7796 | 0.7464 | 0.4076 | 0.7763 | 0.7430 | 0.4099 | 0.7714 | 0.7424 | 0.4127 | 0.7658 |
| DisenGCD-RCD | 0.7548 | 0.4045 | 0.7874 | 0.7507 | 0.4066 | 0.7828 | 0.7482 | 0.4073 | 0.7795 | 0.7470 | 0.4098 | 0.7767 |
| **DisenGCD** | **0.7582** | **0.4004** | **0.7932** | **0.7527** | **0.4039** | **0.7867** | **0.7513** | **0.4052** | **0.7832** | **0.7479** | **0.4076** | **0.7802** |

presents the results of DisenGCD and RCD obtained under different proportions of noise: the orange polyline denotes DsienGCD's results and the blue polyline denotes RCD's results.

As can be seen, the proposed DisenGCD always performs better than RCD in both datasets under different noise ratios. Besides, as the noise ratio increases, the performance leading of the proposed DisenGCD to RCD becomes more significant. That further demonstrates the proposed DisenGCD is more robust against student interaction noise and thus validates the effectiveness of the proposed disentangled graph learning framework.

## B.3 Validation of the Devised Diagnostic Function

In this paper, in addition to the disentangled learning framework and the devised meta multigraph module, we also designed a novel diagnostic function to adopt them. To validate its effectiveness, we compared it with the diagnostic functions of MIRT, NCD, and RCD on the Math dataset, where four splitting settings were considered. Table 8 summarizes their overall performance regarding AUC, ACC, and RMSE.

We can get the following two observations. Firstly, the comparisons of MIRT and DisenGCD-MIRT, NCD and DisenGCD-NCD, as well as RCD and DisenGCD-RCD show that the representations learned by the proposed DisenGCD are effective. Secondly, the performance superiority of DisenGCD to other variants indicates the devised diagnostic function is effective. To sum up, the effectiveness of the devised diagnostic function is validated.

## B.4 Hyperarameter Sensitivity Analysis

For the proposed DisenGCD, there is an important hyperparameter, i.e., the number of hyper-nodes in the meta multigraph. To investigate its influence on theDisenGCD, the hyper-node number was set to 4, 5, 6, and 7, respectively, and we executed the experiments on the ASSISTments and SLP datasets. Figures 6 and 7 plot the results regarding AUC and RMSE.

As can be seen in both two datasets, the DisenGCD can achieve optimal results in terms of RMSE and AUC, when the number of hyper-nodes is set to 5. When the number of hyper-nodes is less than 5, the meta multigraph contains too few propagation paths, adversely affecting the learning of student representations. This results in suboptimal performance compared to the scenario where the number of hyper-nodes is 5. Conversely, when the number of hyper-nodes is greater than 5, there is an abundance of propagation paths, making the aggregated information overly complex and hindering effective student representation learning. Besides, more hyper-nodes cause more computational complexity. Therefore, this paper set the number of hyper-nodes to 5 for the proposed disenGCD.

Furthermore, we aim to explore the influence of the number of hyper-nodes on the robustness of the proposed DisenGCD. Therefore, we also executed the experiments on Math datasets under different ratios of noise interaction(1%,30%,50%). Figures 8 plot the results regarding AUC and ACC. As can be seen, when the number of hyper nodes is equal to 5, the robustness of DisenGCD is the most promising, whose performance under different ratios of noises is balanced better than RCD.

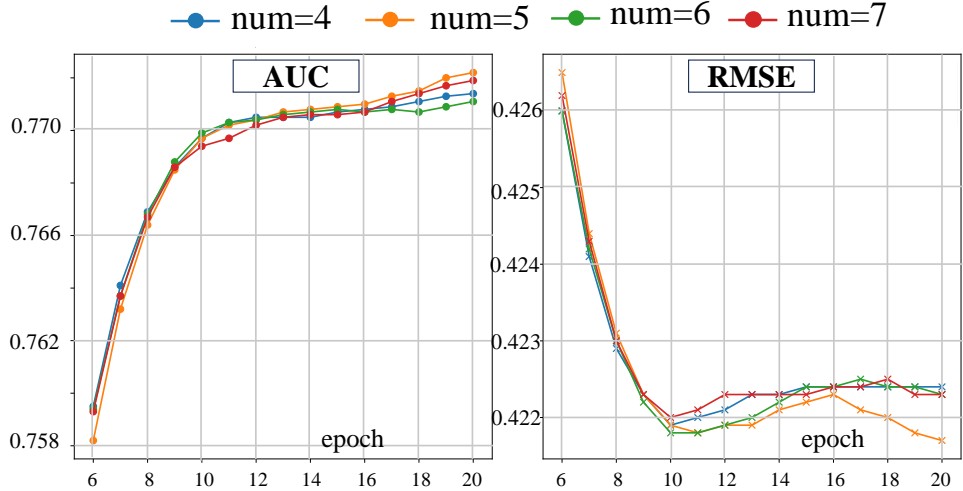

Figure 6: The impact of the number of hyper nodes in the meta-multigraph on the ASSISTments dataset

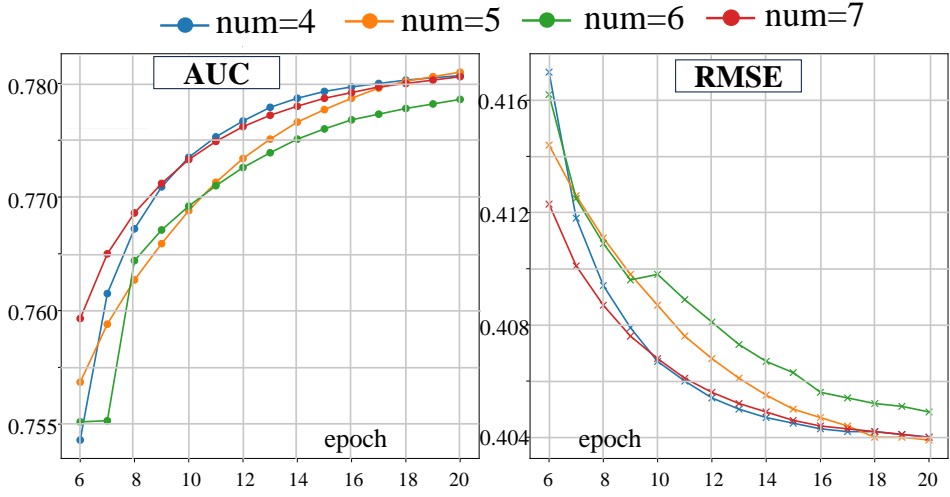

Figure 7: The impact of the number of hyper nodes in the meta-multigraph on the Math dataset

## B.5 Efficiency Analysis of DisenGCD

To investigate the computational efficiency of the proposed approach, we have compared it with RCD on ASSISTments and Math datasets in terms of model inference time and training time. Table 9 presents the overall comparison, where the time of the proposed DisenGCD under different hyperparameters P is also reported.

As can be seen, DisenGCD's inference time is better than RCD's. This indicates that the proposed DisenGCD is more efficient than RCD, further signifying that it is promising to extend DisenGCD to

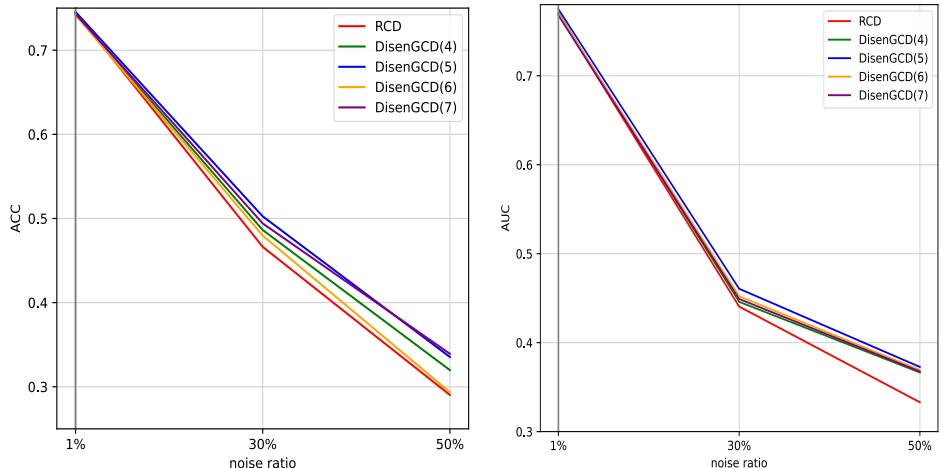

Figure 8: Robustness of DisenGCD under different $P$ to different ratios of interaction noise (on the Math dataset).

| Datasets | Models | Inference time (seconds) |
|---|---|---|
| ASSISTments | RCD | 0.07829 |
| | DisenGCD | 0.01661 |
| Math | RCD | 0.10888 |
| | DisenGCD | 0.00791 |

| Models | ASSISTments | | | Math | | |
|---|---|---|---|---|---|---|
| | RMSE | AUC | Training time(s) | RMSE | AUC | Training time(s) |
| RCD | 0.4245 | 0.7687 | 27161 | 0.4078 | 0.7806 | 22164 |
| DisenGCD(4) | 0.4224 | 0.7714 | 17880 | 0.404 | 0.7857 | 3045 |
| DisenGCD(5) | 0.4217 | 0.7722 | 18135 | 0.4039 | 0.786 | 3346 |
| DisenGCD(6) | 0.4223 | 0.7711 | 18924 | 0.4049 | 0.7836 | 3675 |
| DisenGCD(6) | 0.4223 | 0.7719 | 20004 | 0.404 | 0.7856 | 4681 |

Table 9: **Upper**: Inference time comparison between RCD and DisenGCD. **Lower**: Performance and computational efficiency (regarding training runtime seconds) of DisenGCD under different $P$ (equal to 4, 5, ,6, and 7).

dynamic CD. It can be seen from Table III:Lower: although a larger P will make DisenGCD take more time to train the model, the proposed DisenGCD achieves the best performance on both two datasets when P=5 and its runtime does not increase too much, which is much better than RCD.

| Method/Metric | DisenGCD(GCN) | DisenGCD(GraphSage) | DisenGCD(HAN) | DisenGCD |
|---|---|---|---|---|
| ACC | 0.7428 | 0.7543 | 0.7435 | **0.7582** |
| RMSE | 0.4112 | 0.402 | 0.411 | **0.4004** |
| AUC | 0.7639 | 0.7881 | 0.7647 | **0.7932** |

Table 10: Performance comparison of variants of DisenGCD using different GNNs, and DisenGCD on the Math dataset

### B.6 Ablation Experiments on Graph Representation Learning

To verify the superiority of the employed GAT on the proposed DisenGCD, three GNNs (GCN, GraphSage, and HAN) are used to replace the GAT in the proposed DisenGCD, which are termed DisenGCD(GCN), DisenGCD(GraphSage), and DisenGCD(HAN), respectively. Then, we compared the three approaches with the proposed DisenGCD on the Math dataset. The comparison of them is presented in Table 10 As shown in Table 10, the GCN-based DisenGCD achieves the worst performance, followed by DisenGCD(HAN). The GraphSage-based DisenGCD holds a competitive performance to yet is still worse than the proposed DisenGCD (based on GAT).

The above results show the GAT is a suitable and optimal choice among these four GNNs for the proposed DisenGCD, but we think the GAT may not be the best choice because there exist other types of GNNs that can be integrated with DisenGCD, showing better performance.

In summary, it is reasonable and effective for the proposed DisenGCD to adopt the GAT to learn the representations.

## Appendix C   Limitation Discussion

There are still some limitations in the proposed DisenGCD.

- **High Computational Complexity and Poor Scalability.** The complexity of DisenGCD consists of three parts: the aggregation of $P$-hypernode meta multigraph on $N + M + K$-node interaction graph (average $A_1$ neighbors), L-layer GAT aggregation on the $M + K$-node relation graph (average $A_2$ neighbors), and L-layer GAT aggregation on the $K$-node dependency graph (average $A_3$ neighbors). Suppose each node is $d$ dimensional, L-layer GAT on $K$-node graph contains: computing nodes' attention from neighbors $O(L \times K \times A_3 \times d)$ and nodes' linear transformation $O(L \times K \times d^2)$, totally equalling $O(L \times K \times d \times (A_3 + d))$; meta multigraph on $N + M + K$-node graph contains: computing nodes' attention $O(P \times (N + M + K) \times A_1 \times d)$, computing path's attention $O(P \times (N + M + K) \times d)$, and nodes' linear transformation $O(P \times (N + M + K) \times d^2)$. As a result, DisenGCD's complexity equals $O(P(N+M+K)d(A_1+1+d))+O(L(M+K) \times d(A_2+d))+O(L \times K \times d(A_3+d))$. As analyzed above, the time complexity of our model may be high and not applicable in some large data sets.

- **Potentially Poor Task Transferability.** The proposed method primarily targets cognitive diagnosis tasks and is designed to handle such graphs as the *student-exercise-concept* graph, which may not be readily applicable to other tasks. It is specifically tailored for modeling tasks similar to the student-exercise-concept triad diagram.

