# OpenReview forum: "DisenGCD: A Meta Multigraph-assisted Disentangled Graph Learning Framework for  Cognitive Diagnosis"
_NeurIPS.cc/2024/Conference — NeurIPS 2024 poster_

### Official Review · Reviewer_53z4 · 2024-07-10

**Soundness:** 3
**Presentation:** 2
**Contribution:** 1
**Rating:** 5
**Confidence:** 4

**Summary:**

The paper studies cognitive diagnosis based on graph learning. The proposed method utilizes multiple graphs to assist the representation learning. Specifically, the authors further disentangle another two graphs from the most comprehensive student-exercise-concept interaction graph by removing certain node and edge types. Therefore, three kinds of disentangled representations are learned based on the three graphs. Lastly, a diagnostic function fuses the three representations for final predictions.

**Strengths:**

1. The problem that the paper aims to solve is well formulated and it is of practical value.
2. The authors give very detailed illustrations of the methodology part.
3. Comprehensive experiments are conducted to demonstrate the effectiveness of the proposed method.

**Weaknesses:**

1. Since the proposed method learns the graph representations based on disentangled graphs, the authors are encouraged to add a section about disentangled graph representation learning in the related work part to summarize the recent works on this problem and illustrate the relation of this work with previous ones.
2. A more detailed introduction to incorporated datasets should be given. Specifically, both accuracy and RMSE are employed as evaluation metrics, the authors should provide a clearer illustration of the target tasks of those datasets.
3. The authors should include more mode variants to demonstrate the effectiveness of the added graphs (e.g., w/o G_R, w/o G_D, and w/o both).
4. GAT is used in this work to learn exercise and concept representations. The authors are encouraged to introduce the reason for this choice or conduct experiments based on other graph model backbones.
5. There are many typos in the paper, e.g., line 16 and line 254. The authors should check the grammar carefully.

**Questions:**

Please refer to the weakness part.

---

> ### Author Rebuttal · Authors · 2024-08-07
>
> ## Response to Q.1
>
> The related work on disentangling graph representation learning is as follows, and we will add this to the revised paper:
>
> 1. Learning potential representations of disentangling in complex graph networks to achieve model robustness and interpretability has been a hot topic in recent years. Researchers have put forward a lot of classic neural network disentanglement approaches (e.g.,  DisenGCN[R1], DisenHAN[R2], DGCF[R3], DCCF[R4], DcRec[R5], etc.) to address this challenge. Some representative approaches will be briefly introduced.
>
> 2. For example, in DisenHAN[R1], the authors utilized disentangled representation learning to account for the influence of each factor in an item. They achieved this by mapping the representation into different spatial dimensions and aggregating item information from various edge types within the graph neural network to extract features from different aspects； In DCCF[R4], the authors introduced global intent disentanglement into graph contrastive learning, extracting more fine-grained latent factors from self-supervised signals to enhance model robustness； DcRec[R5] disentangles the network into a user-item domain and a user-user social domain, generating two views through data augmentation, and ultimately obtaining a more robust representation via contrastive learning.
>
> 3. Despite many approaches suggested, they were primarily applied to bipartite graphs to learn different representations from different perspectives, for learning more comprehensive representations. While  this paper aims to leverage disentanglement learning to mitigate the influence of the interaction noise in the interaction graph, and thus we proposed a meta multigraph-assisted disentangled graph cognitive diagnostic framework to learn three types of representations on three disentangled graphs. By doing so, the influence of the noise on exercise and concept learning can be well alleviated.
>
> ```
> [R1]Jianxin Ma, et al., Disentangled Graph Convolutional Networks.
> [R2]Yifan Wang, et al., DisenHAN: Disentangled Heterogeneous Graph Attention Network for Recommendation.
> [R3]Xiang Wang, et al., Disentangled Graph Collaborative Filtering.
> [R4]Xubin Ren, et al., Disentangled Contrastive Collaborative Filtering.
> [R5]Jiahao Wu, et al., Disentangled Contrastive Learning for Social Recommendation
> ```
>
> ## Response to Q.2
>
> We would like to  answer your question from the following three aspects:
>
> - In the Appendix, we indeed provided a brief introduction to the three datasets (ASSISTments, Math, SLP). However, this introduction may not be detailed, thus we will give more information about the datasets in the revised paper, including their collectors, the contained subjects, and many other attributes.
> - The cognitive diagnosis task is to assess the true knowledge level of students. To achieve this, this task is generally solved by modeling the students' answer prediction task, which is a binary classification task. Therefore, we use the evaluation metrics of classification to measure the diagnosis accuracy.
> - The utilized metrics include **ACC**, **RMSE**, and **AUC**, which are widely used for the classification tasks. The three metrics are also widely adopted in many previous CD approaches.
>
> ## Response to Weak.3
>
> As you suggested, we added three variants of the DisenGCD to show the effectiveness of the added graphs, which are as
> - DisenGCD(w/o both) is same as DisenGCD but learns exercise and concept representations directly through **native embedding modules(NEMs)** .
> - DisenGCD(w/o $\mathcal{G_R}$)  is same as DisenGCD but its exercise representation is learned through  a **NEM**.
> - DisenGCD(w/o $\mathcal{G_D}$) is same as DisenGCD but its concept representation is learned through a **NEM**.
>
> **Table II** in ***global.pdf*** summarizes the results of all variants and the DisenGCD on the ASSISTments dataset.
>
> As can be seen, the DisenGCD outperforms both DisenGCD(w/o $\mathcal{G_D}$) and DisenGCD(w/o $\mathcal{G_R}$), which indicates the effectiveness of learning exercise/concept representations by the  GAT on added graphs. Moreover, the performance leading of DisenGCD(w/o $\mathcal{G_D}$) and DisenGCD(w/o $\mathcal{G_R}$) over DisenGCD(w/o both) can further validate the effectiveness of the added graphs. In addition, the effectiveness of the DisenGCD can be indirectly validated through the comparisons of DisenGCD(w/o both) and DisenGCD(I), as well as DisenGCD(w/o $\mathcal{G_D}$) and DisenGCD(Is+Rec).
>
> In summary, the above comparison validates the effectiveness of the added graphs and the proposed disentangling learning framework.
>
> ## Response to Q.4
> As you suggested, other three types ofGNNs (i.e., GCN, GraphSage, and HAN) are used as the graph model backbones to replace the GAT module in DisenGCD. The variants of DisenGCD, utilizing GCN, GraphSage, and HAN as the backbones, are denoted as DisenGCD(GCN), DisenGCD(GraphSage), and DisenGCD(HAN).
>
> To observe their influence on the proposed approach, these three variants were executed on the Math dataset, whereas the other two datasets were not used due to the time limit. As a result,  **Table I:Upper** in ***global.pdf*** summarizes the results of these variants and the DisenGCD. We can find that the variant DisenGCD(GraphSage) can obtain competitive performance than state-of-the-art approaches, while DisenGCD(GCN) and DisenGCD(HAN) cannot. Despite its competitive performance, DisenGCD(GraphSage) is still worse than DisenGCD, which indicates that the GAT is most suitable for the proposed framework.
>
> In summary, the results show the GAT is a suitable and optimal choice among these four GNNs for the proposed DisenGCD, and it is reasonable and effective for the proposed DisenGCD to adopt the GAT to learn the representations.
>
> ## Response to Q.5
> Thanks for pointing out the topos.  We will correct these typos and double-check the paper.
>
> ## We appreciate your valuable feedback and will revise the paper based on the above.

---

> > ### Comment · Reviewer_53z4 · 2024-08-10
> >
> > Thanks for the detailed response from the authors. After reading the rebuttal, I decide to raise my score to 5.

---

> ### Author Response · Authors · 2024-08-11
>
> Thank you for your positive feedback. We greately appreciate your recognition of our efforts to address your concerns.

---

### Official Review · Reviewer_7tDb · 2024-07-11

**Soundness:** 3
**Presentation:** 3
**Contribution:** 3
**Rating:** 7
**Confidence:** 4

**Summary:**

In this paper, the authors introduce a meta multigraph assisted disentangled graph cognitive diagnosis model. Its main contribution is to propose a disentangled graph framework, which disentangles the student-exercise-concept dependency graph into an exercise-concept interaction graph and a concept-concept relationship graph to improve the robustness of representation modeling. Additionally, the concept of meta multigraph is introduced in the student-exercise-concept dependency graph to assist in better modeling of student representation. DisenGCD has demonstrated good performance compared to previous models in multiple datasets, and has shown good robustness in the presence of noise interference or sparse datasets. Finally, the authors also discussed some limitations of DisenGCD in terms of computational complexity and task transferability.

**Strengths:**

1.	The paper is well-organized and clearly written, making it accessible to readers. Besides, the authors also present mathematical formulations, detailed explanations of the proposed models.
2.	The authors focus on improving the robustness of the CD model and provide a detailed analysis of some problems found in previous graph cognitive diagnosis works.
3.	The experiments are very thorough. The authors demonstrate the robustness of the model through decoupling experiments, sparse dataset experiments, and noisy dataset experiments

**Weaknesses:**

1. Although DisenGCD has demonstrated excellent performance across multiple datasets, it still appears to have the drawback of high computational complexity, which may limit its application in practical scenarios.
2. Although the model has shown improved performance, its interpretability may be lacking. In real-world applications within the education field, transparent and interpretable models are more likely to gain acceptance and trust. However, the paper does not address methods for enhancing the interpretability of the model.
3. The paper lacks detailed discussion on the challenges and potential issues that may arise when applying the model in practical educational settings. For instance, it does not address the strategies for handling data privacy and security concerns, which are critical considerations in the implementation of such models.

**Questions:**

1. KaNCD seems to have been found to work equally well in some papers. Can you compare DisenGCD and KaNCD?
2. The paper utilizes graph attention networks (GAT) to learn representations of exercises and concepts. However, it would be valuable to explore and compare alternative graph neural network (GNN) models, such as graph convolutional networks (GCN) or LightGCN, to assess their performance in the same context.
3. How are the multiple learnable propagation paths in the meta-multigraph learning module designed and selected? Have you tried different numbers or types of propagation paths and how well have they worked?

**Limitations:**

\

---

> ### Author Rebuttal · Authors · 2024-08-07
>
> ## Response to Q.1
>
>
> As you suggested, we added KaNCD [R1] to be compared with the proposed DisenGCD, and the comparison results are summarized in  **Table I:Upper** in ***global.pdf***, where only the Math dataset is used because there is no enough time to validate on other two larger datasets.
>
> As you can see, in addition to KaNCD, we have also compared our approach with  SCD [R2] and KSCD [R3], which are other two recently published CDMs. As shown in  **Table I:Upper** in ***global.pdf***, the proposed DisenGCD still performs better than KaNCD and other two state-of-the-art CDMs, further validating the effectiveness of the proposed DisenGCD.
>
> In the future revised paper, we will add these CDMs to the comparison on all three datasets.
>
> ```
> [R1] F. Wang, Q. Liu, E. Chen, Z. Huang, Y. Yin, S. Wang, and Y. Su, “Neuralcd: a general framework for cognitive diagnosis,” IEEE Transactions on Knowledge and Data Engineering, 2022.
> [R2] Wang S, Zeng Z, Yang X, et al. Self-supervised graph learning for long-tailed cognitive diagnosis[C]//Proceedings of the AAAI conference on artificial intelligence. 2023, 37(1): 110-118.
> [R3] H. Ma, M. Li, L. Wu, H. Zhang, Y. Cao, X. Zhang, and X. Zhao, “Knowledge-sensed cognitive diagnosis for intelligent education platforms,” in Proceedings of the 31st ACM International Conference on Information & Knowledge Management, 2022, pp. 1451–1460.
> ```
>
>
> ## Response to Q.2
>
> To solve your concern, three GNNs (GCN, GraphSage, and HAN) are used to replace the GAT in the proposed DisenGCD, which are termed DisenGCD(GCN), DisenGCD(GraphSage), and DisenGCD(HAN), respectively. Then, we compared the three approaches with the proposed DisenGCD on the Math dataset, where the other two datasets were not used due to the time limit. The comparison of them is presented in **Table I:Upper** in ***global.pdf***.
>
>
> As shown in **Table I:Upper**, the GCN-based DisenGCD achieves the worst performance, followed by DisenGCD(HAN). The GraphSage-based DisenGCD holds a competitive performance to yet is still worse than the proposed DisenGCD (based on GAT).
>
> The above results show the GAT is a suitable and optimal choice among these four GNNs for the proposed DisenGCD, but we think the GAT may be not the best choice because there exist other types of GNNs that can be integrated with DisenGCD, showing better performance.
>
> In summary, it is reasonable and effective for the proposed DisenGCD to adopt the GAT to learn the representations.
>
> ## Response to Q.3
>
> The design of the learnable propagation paths  is inspired by the success of meta graph. And the propagation paths are selected by the adopted routing strategy, which learns a value for each candidate propagation path. Afterward, these propagation paths that satisfy the threshold will be automatically kept. Therefore, the selection of propagation paths is achieved by the model learning.
>
>
> As for the influence of the propagation path number on the proposed approach, it can be found in line 179 that the path number is determined by the number of hyper nodes (i.e., *P*). Therefore, this influence can be investigated by analyzing the influence of the number of hyper nodes (i.e., *P*).
>
> In fact, we have investigated the influence of  *P* on the proposed DisenGCD in ***Appendix B.4***. As shown in Figures 6 and 7 in Appendix B.4,  the proposed DisenGCD is not very sensitive to the number of hyper nodes, where DisenGCD obtains a promising performance when *P* is 5. That indirectly indicates that the proposed  DisenGCD is not sensitive to the number of candidate propagation paths.
>
> In addition, for more convincing, the influence of different P on the robustness of the proposed DisenGCD is also analyzed as shown in **Fig.1(a)&(b)** in ***global.pdf***, where only the Math dataset is used due to the time limit. **Fig.1(a)&(b)** present the results of RCD and DisenGCD with different P obtained under different ratios of noise interactions (1%, 30%, and 50%). As can be seen, the robustness of DisenGCD is a bit sensitive to different *P*, and the DisenGCD achieves the most promising robustness when *P* is equal to 5. That indirectly indicates that the proposed  DisenGCD is sensitive to the number of candidate propagation paths.
>
> In summary, the performance of the proposed DisenGCD is not sensitive to the propagation path number while its robustness is sensitive. Nevertheless, the proposed DisenGCD can obtain promising performance and robustness when *P* is set to 5.
>
>
>
>
> ## We appreciate your valuable feedback and will revise the paper based on the above.

---

> > ### Comment · Reviewer_7tDb · 2024-08-13
> >
> > The author has addressed some of my concerns, and I am inclined to raise my score. However, some issues still need further improvement in the final version.

---

> ### Author Response · Authors · 2024-08-13
>
> Thank you for your quick feedback and for contributing to our improved score. We greatly appreciate your efforts. We apologize that some issues still need improvement, and we would like to resolve them as follows.
>
> ## Response to Weakness 1
> To investigate the computational efficiency of the proposed approach, we have compared it with RCD on ASSISTments and Math datasets in terms of model inference time and training time. **Table III** in ***global.pdf*** presents the overall comparison, where the time of the proposed DisenGCD under different hyperparameters *P* is also reported.
>
> As can be seen, DisenGCD's inference time is better than RCD's. This indicates that the proposed DisenGCD is more efficient than RCD, further signifying that it is promising to extend DisenGCD to dynamic CD.
> It can be seen from **Table III:Lower**: although a larger *P* will make DisenGCD take more time to train the model,  the proposed DisenGCD achieves the best performance on both two datasets when *P*=5 and its runtime does not increase too much, which is much better than RCD.
>
> In summary, the above comparison shows the model efficiency superiority of the proposed DisenGCD.
>
> ## Response to Weakness 2
>
> We have to argue that the diagnostic function in the proposed DisenGCD is interpretable.
>
>
>
> Firstly, $\mathbf{h}\_{si} = F_{si}(\overline{\mathbf{S}_i}+ \overline{\mathbf{C}_k})$ denotes the combination of the learned student representation and the learned concept representations, which is similar to the combination in RCD, i.e., $Concat(\mathbf{S}\_i, \mathbf{C}_k)$. As a result, $\mathbf{h}\_{si}$ can be seen as the  student's mastery of each knowledge concept.
>
> Secondly  $\mathbf{h}\_{ej} = F\_{ej}(\overline{\mathbf{E}_j}+\overline{\mathbf{C}_k})$ denotes the combination of the learned exercise representation and the learned concept representations, which is also similar to the combination in RCD for exercises, i.e., $Concat(\mathbf{E}\_j, \mathbf{C}_k)$. Therefore, $\mathbf{h}\_{ej}$ can represent  the exercise difficulty of each concept.
>
> Thirdly, $\mathbf{h}\_{simi} = \sigma(F\_{simi}(\mathbf{h}\_{si}\cdot\mathbf{h}\_{ej}  ))$ is mainly  used to measure the similarity between $\mathbf{h}\_{si}$ and $\mathbf{h}\_{ej}$ via a dot-product followed by an FC layer and a Sigmoid function.  Therefore, a higher value in each bit of $\mathbf{h}\_{simi}$ represents a higher mastery on each concept, further indicating a higher probability of answering the related exercises.
>
> Finally, $\hat{r\_{ij}} = (\sum Q_k\cdot \mathbf{h}_{simi})/\sum Q_k$ has a similar idea to NCD to compute the overall mastery averaged over all concepts contained in exercise $e_j$.
>
> In summary, the proposed model is well-interpretable, as agreed by **Reviewer 2m1M**.
>
>
> ## Response to Weakness 3
>
> We have to admit that data privacy and security are indeed a crucial problem in intelligent education.
>
> However, existing cognitive diagnosis models did not explore this. To the best of our knowledge, we think the following two potential techniques can be applied to CD to solve data privacy and security issues.
>
> 1. The first is to apply the differential privacy technique.  Differential privacy [R1] protects individual data by adding noise, ensuring personal information can’t be easily inferred from data analysis. Differential privacy is widely used in fields like computer vision [R2], recommender systems [R3,R4], and many others [R5].
>
> Among these, the recommendation task is most related to cognitive diagnosis (CD), and thus the differential privacy can be easily extended to CD by summarizing the experiences of these approaches. A successful example of extending techniques in the recommendation to CD is the MF [R6] model.
>
> 2. The second is to apply the Federated Learning technique. It trains a model across multiple devices without sharing data, improving privacy and efficiency. It has succeeded in many domains, including recommender systems [R7].
>
> According to its successful experiences in the recommendation, federated learning can also be extended to CD. Moreover, a few studies [R8] have applied federated learning to knowledge tracing (KT), making good progress. KT is the most related task to CD in intelligent education; thus, their experiences can be brought to CD to solve data privacy and security problems.
>
> To sum up, the proposed DisenGCD and existing CDMs could be combined with differential privacy and federated learning to solve the data privacy and security problem in intelligent education.  We will discuss this problem in revised paper.
> ```
> [R1] Dwork, Differential privacy.
> [R2] Mixed differential privacy in computer vision.
> [R3] A differential privacy framework for matrix factorization recommender systems.
> [R4] Applying differential privacy to matrix factorization.
> [R5] Differential privacy: A survey of results.
> [R6] Matrix factorization techniques for recommender systems.
> [R7] Federated recommendation systems.
> [R8] Federated deep knowledge tracing.
> ```

---

### Official Review · Reviewer_9V3d · 2024-07-12

**Soundness:** 3
**Presentation:** 3
**Contribution:** 2
**Rating:** 5
**Confidence:** 3

**Summary:**

This paper introduces DisenGCD, a new framework for cognitive diagnosis in educational contexts. The authors make several contributions:
1) They propose a disentangled graph learning approach, separating the typically unified graph into three distinct graphs: student-exercise-concept interactions, exercise-concept relations, and concept dependencies.
2) They develop a meta multigraph module for learning student representations, which innovatively allows access to lower-order exercise representations.
3) They employ GAT-based modules to learn exercise and concept representations on the disentangled graphs.
4) They design a new diagnostic function to effectively combine the learned representations from the different graphs.
The authors evaluate DisenGCD on multiple datasets, demonstrating performance improvements and enhanced robustness compared to state-of-the-art methods in cognitive diagnosis.

**Strengths:**

The paper presents several notable strengths:
1) The proposed disentangled graph learning approach is a novel contribution to the field of cognitive diagnosis. By separating different types of information (student-exercise-concept interactions, exercise-concept relations, and concept dependencies) into distinct graphs, the authors have created a more robust model that appears less susceptible to noise in the data. This is a significant advancement over existing unified graph approaches.
2) The meta multigraph module introduces an innovative mechanism for student representation learning. By allowing access to lower-order exercise representations, this module potentially captures more nuanced relationships in the data, which could be particularly valuable in educational contexts where student knowledge evolves over time.
3) The empirical evaluation is comprehensive and well-executed. The authors conduct experiments on multiple datasets (ASSISTments, Math, and SLP), demonstrating consistent improvements in performance and robustness over state-of-the-art methods. This thorough evaluation strengthens the paper's claims and increases confidence in the generalizability of the proposed method.
4) The paper includes detailed ablation studies and analyses that effectively validate the different components of DisenGCD. These studies provide valuable insights into the contribution of each component and help justify the design choices made by the authors.

**Weaknesses:**

1) While the disentangled graph approach is innovative, it may lose some valuable cross-entity interactions. The separation of student-exercise interactions from exercise-concept relations could potentially limit the model's ability to capture complex, multi-hop relationships that might exist in a unified graph structure.
2) The meta multigraph module, although showing improvements, adds considerable complexity to the model. It's not clear if this added complexity is always justified, particularly for simpler datasets or scenarios with limited noise. A more thorough analysis of the trade-off between model complexity and performance gains would strengthen the paper.
3) The paper lacks a comprehensive discussion on how DisenGCD handles cold-start problems, especially for new students or exercises that don't have established interactions in the graph structure. This is a common challenge in educational settings and needs to be discussed.

**Questions:**

1) How does DisenGCD adapt to dynamic changes in the knowledge structure, such as the introduction of new concepts or the discovery of new relationships between existing concepts? Does the disentangled structure present any challenges in updating the model under these scenarios?
2) In Section 4.1, you describe the meta multigraph as containing P hyper-nodes, and in your experiments you set P to 5. How did you determine this value? Have you explored the impact of different values of P on the model's performance and computational efficiency?
3) While your experiments demonstrate improved robustness to interaction noise, how does DisenGCD perform when noise is introduced in the concept dependency graph or exercise-concept relation graph? This could simulate errors in curriculum design or expert knowledge.
4) Have you investigated whether the disentangled approach leads to any loss of performance in scenarios where complex, cross-entity relationships are crucial for accurate diagnosis? Are there cases where a unified graph approach might outperform DisenGCD?

---

> ### Author Rebuttal · Authors · 2024-08-07
>
> ## Response to Q.1
> We must admit that the design of the proposed DisenGCD did not consider the scenery of dynamic changes in knowledge structures. However, the proposed DisenGCD and RCD are trained in an inductive manner in graph learning. Therefore, it is feasible for DisenGCD and RCD to handle the scenery of introducing new concepts, but their performance is unknown.
>
> To investigate this, we conducted the experiments on the Math dataset due to the time limit. Specifically, interactions and exercises related to 80% of knowledge concepts were not masked during the training, and we tested the model on the remaining interactions and exercises. The comparison results of the DisenGCD and RCD are presented in **Table I:Lower** in   ***global.pdf***.
>
> We can see DisenGCD exhibits poor performance on such this scenery, which is poorer than RCD. That indicates that the proposed DisenGCD cannot adapt to dynamic changes in the knowledge structure.
>
> The reason may be that the propagation paths learned in the meta-multigraph are not suitable for newly added knowledge concepts.  Since adding 20% of concepts changes the topology of the interaction graph, while different-topology graphs generally need to learn different propagation paths. This fact can be found in Figure 4(a) in the submitted paper, stated in [R1].
>
> In short, the proposed DisenGCD may not adapt to the scenery of introducing new concepts well due to the utilized learnable meta multigraph. In future work, we would like to explore how to address this issue to handle dynamic changes in knowledge structures.
>
> ```
> [R1] Chao Li, Hao Xu, and Kun He. Differentiable meta multigraph search with partial message propagation on heterogeneous information networks.
> ```
> ## Response to Q.2
>
> Actually, setting *P* to 5 is by trial and error on the DisenGCD. In ***Appendix B.4***, we have discussed the influence of different *P* on the performance of DisenGCD. **Figures 6 and 7** plot the results of DisenGCD obtained under different *P* (4, 5, 6, and 7) on the two datasets. It can be found that  DisenGCD can achieve the optimal performance on both two datasets when setting *P* to 5.
>
> However, we did not explore its influence on the computational efficiency.  The most intuitive is that a larger *P* will cause worse computational efficiency.
>
> To investigate this, **Table III:Lower** in ***global.pdf*** presents the performance and runtimes (for training) of RCD and DisenGCD under different *P*. As can be seen, it is consistent with the above assumption that a larger *P* will make DisenGCD take more time to train the model. The proposed DisenGCD achieves the best performance on both two datasets when *P*=5, and the runtime increase from  *P*=4 to *P*=5 is not too much and is acceptable.
> Besides, **Fig.1(a)&(b)** in ***global.pdf***  show that the proposed DisenGCD can obtain the most promising robustness when *P*=5.
>
> Therefore, in this paper, we set *P* to 5 for a comprehensive consideration of model performance, model robustness, and model efficiency.
>
>
> ## Response to Q.3
>
> To figure out the robustness of the proposed DisenGCD to exercise-concept noise, we conducted experiments on the Math dataset, where 1\%, 10\%, 20\%, and 30\% of exercise-concept relations are randomly removed as or added as noise relations, respectively. (Note that larger dataset) **Fig. 1(c)&(d)** in ***global.pdf*** compares the performance of RCD and DisenGCD on Math under different ratios of exercise-concept noise.
>
> As can be seen, it is unexpected that the proposed DisenGCD exhibits better robustness than RCD to the added exercise-concept noises. However, different from against interaction noise (as shown in Figure 3(a) in the submitted manuscript), the robustness superiority/leading of DisenGCD over RCD does not increase as the ratio of noise increases, where the leading performance becomes smaller when a larger portion of noises are added.
>
> It is reasonable because the added exercise-concept noise affects the learning on two graphs (the interaction graph and the relation graph), which further affects the learning of student and exercise representations. Since the learning of the concept representation is not influenced, DisenGCD still outperforms RCD. However, when more noise is added, DisenGCD's robustness against exercise-concept noise will not be better than that of DisenGCD against interaction noise, where the learning of two graphs is affected.
>
>
> ## Response to Q.4
>
> As for the first sub-question, to be honest, we have not investigated the performance of the proposed approach in scenarios where complex, cross-entity relationships. To answer your concern, we have tried our best to check many educational datasets， but we indeed cannot find a suitable dataset that holds cross-entity relations. Therefore, we are sorry that we cannot answer your question through experiments.  However, as far as we are concerned, the DisenGCD can extend to handle cross-entity relations by disentangling more graphs. That may make promising performance, but its complexity and efficiency will be criticized.
> We also would like to answer your question if you can recommend a specific dataset.
>
> As for the second, we would like to state that the compared RCD approach is essentially a unified graph approach, where all representations are learned and exchanged in an implicit graph. As shown in the submitted manuscript, the proposed DisenGCD exhibits better performance than the RCD. In addition, we have also compared  DisenGCD with another unified graph-based approach, i.e., SCD [R1], on the Math dataset, where the results are presented in **Table I:Upper** in ***global.pdf***. We can find that the DisenGCD still outperforms SCD. In summary, there may exist other novel unified graph approaches outperforming DisenGCD, but existing unified graph-based CDMs (RCD and SCD) cannot.
>
> ```
> [R1] Wang S, Zeng Z, Yang X, et al. Self-supervised graph learning for long-tailed cognitive diagnosis[C].
> ```

---

> > ### Comment · Reviewer_9V3d · 2024-08-12
> >
> > In general, I am satisfied with the answers. I also think the paper studies an interesting domain where GNNs are applied. I will keep my score, but the paper seems to be above the acceptance bar.

---

> > > ### Author Response · Authors · 2024-08-13
> > >
> > > Many thanks for taking the time to carefully read the rebuttal. We appreciate your recognition of the value of our work! We are pleased to address your concerns and will revise the paper according to your suggestions.

---

### Official Review · Reviewer_2m1M · 2024-07-13

**Soundness:** 3
**Presentation:** 3
**Contribution:** 3
**Rating:** 5
**Confidence:** 1

**Summary:**

The paper presents DisenGCD, a novel cognitive diagnosis model designed to enhance the robustness and accuracy of student, exercise, and concept representations by leveraging a meta multigraph-assisted disentangled graph learning framework. DisenGCD constructs and disentangles three specific graphs for interactions, relations, and dependencies, thus mitigating the negative impact of noise in students’ interactions. The framework incorporates a meta multigraph learning module and GAT to improve the learning of latent representations, demonstrating superior performance and robustness compared to state-of-the-art CDMs on multiple datasets.

**Strengths:**

1. The use of a meta multigraph-assisted disentangled graph learning framework is a novel approach that significantly enhances the robustness of cognitive diagnosis models.
2. DisenGCD shows better performance in terms of AUC, accuracy, and RMSE compared to other state-of-the-art CDMs.
3. The model effectively handles noise in student interactions, ensuring that the learning process remains accurate and reliable.
4. The experiments cover multiple datasets and different data splits, providing a thorough validation of the model’s effectiveness.
5. The diagnostic function of DisenGCD maintains high interpretability, comparable to traditional models like NCD, IRT, and MIRT.

**Weaknesses:**

1. The implementation of the meta multigraph learning module and the disentangled graph learning framework could be complex and resource-intensive.
2. The paper does not report error bars or other statistical significance measures, which are important for validating experimental results.
3. The comparison does not include some recent models like SCD, which could provide a more comprehensive evaluation.
4. While some limitations are discussed, a more detailed analysis of potential weaknesses and future improvements could enhance the paper’s transparency and reliability.

**Questions:**

1. Can the model be adapted to handle real-time data for dynamic cognitive diagnosis?
2. How does the choice of hyperparameters affect the model’s performance and robustness?
3. Can the disentangled graph learning framework be integrated with other neural network architectures for further performance enhancement?

---

> ### Author Rebuttal · Authors · 2024-08-07
>
> ## Response to   Weak.1 & Q.1
>
> We admit that the proposed DisenGCD seems complex and needs more resources for its implementation because more graphs need to be handled. In future work, we would like to design a more efficient paradigm to learn robust representation in a unified graph.
>
> As for **Question 1**, to be honest, we have to admit that the proposed DisenGCD is not designed for real-time dynamic cognitive diagnosis.
> However, as shown in some research [R1,R2], the graph-based CD  approaches could be easily extended to knowledge tracing (KT, a task similar to dynamic cognitive diagnosis): for example, RKT [R1] is actually a variant of RCD in knowledge tracing, where the representations learned by graph NNs are attached as static exercise features used for sequence prediction.
>
> Due to the time limit and our limited coding capabilities， we are sorry that we failed to extend our approach to dynamic cognitive diagnosis.
>
> However, to extend graph CD approaches to KT or dynamic CD, a crucial problem to be solved is the graph-based model's efficiency. Therefore,  to solve your concern indirectly, we make the statistics about the proposed DisenGCD's efficiency regarding its inference time.  To this end, the inference time (seconds) of  DisenGCD and RCD are compared in **Table III:Upper** in ***global.pdf***.
>
> As can be seen, the inference time of DisenGCD is better than RCD. That indicates that the proposed DisenGCD is more efficient than RCD, further signifying that it is promising to extend DisenGCD to dynamic CD.
>
> ```
> [R1] Pandey, S., & Srivastava, J. (2020, October). RKT: relation-aware self-attention for knowledge tracing. In Proceedings of the 29th ACM international conference on information & knowledge management (pp. 1205-1214).
> [R2] Yang, Y., Shen, J., Qu, Y., Liu, Y., Wang, K., Zhu, Y., ... & Yu, Y. (2021). GIKT: a graph-based interaction model for knowledge tracing. In Machine learning and knowledge discovery in databases.
> ```
>
>
> ## Response to Q.2
> The most important hyperparameter is the number of hyper nodes (i.e., *P*) in the meta-multigraph. To solve your concern, we would like to explore its influence from **two aspects** (i.e., on performance and robustness):
>
> **On the one hand**, its influence on model performance has been discussed in ***Appendix B.4***.  As shown in Figures 6 and 7 in Appendix B.4, the proposed DisenGCD can achieve promising performance when the number of hyper nodes is 5, indicating setting *P* to 5 is reasonable.
>
> ***On the other hand***, we also analyzed the influence of different *P* (set to 4, 5, 6, and 7 ) on the model robustness under the Math dataset (ASSISTments is not used due to the time limit). **Fig.1(a)&(b)** in ***global.pdf***  shows the results of RCD and DisenGCD with different *P* obtained under different ratios of noise interactions (1\%, 30\%, and 50\%). As you can see, when the number of hyper nodes is equal to 5, the robustness of DisenGCD is the most promising, whose performance under different ratios of noises is balanced better than RCD.
>
> In summary, it can be seen that the proposed DisenGCD can achieve good performance and promising robustness when setting the number of hyper nodes to 5.
>
> ## Response to Q.3
>
> Other NNs can be integrated with the proposed framework yet the performance of them are not sure.
>
> To validate this, we used three NNs (GCN, GraphSage, and HAN) to replace the GAT in the proposed approach, respectively, which are denoted as DisenGCD(GCN), DisenGCD(GraphSage), and DisenGCD(HAN). Due to the time limit, we only compared them with the proposed approach on the Math dataset, and the comparison results are summarized in **Table I:Upper** in ***global.pdf***.  It can be seen that the GAT enables the proposed framework to hold the best performance than other three variants, where  DisenGCD(GraphSage) performs better than the other two.
>
> The above comparison indicates that the proposed DisenGCD could be integrated with other neural network architectures and their performance may be improved when suitable NNs are adopted.
>
> ## Response to Weak.3
>
> According to your suggestions, we have added more recent CDMs, like SCD [R1], KaNCD [R2], and KSCD [R3] to the comparison. The comparison between them and DisenGCD is presented in  **Table I:Upper** in ***global.pdf***, where only the Math dataset is used due to the time limit.
>
> As can be seen from  **Table I:Upper**, compared to these recent CDMs, the proposed DisenGCD still exhibits better performance.
>
> We will add these CDMs to the comparison in the revised paper.
>
> ```
> [R1] Wang S, Zeng Z, Yang X, et al. Self-supervised graph learning for long-tailed cognitive diagnosis[C]//Proceedings of the AAAI conference on artificial intelligence. 2023, 37(1): 110-118.
> [R2] F. Wang, Q. Liu, E. Chen, Z. Huang, Y. Yin, S. Wang, and Y. Su, “Neuralcd: a general framework for cognitive diagnosis,” IEEE Transactions on Knowledge and Data Engineering, 2022.
> [R3] H. Ma, M. Li, L. Wu, H. Zhang, Y. Cao, X. Zhang, and X. Zhao, “Knowledge-sensed cognitive diagnosis for intelligent education platforms,” in Proceedings of the 31st ACM International Conference on Information & Knowledge Management, 2022, pp. 1451–1460.
> ```
>
> ## We appreciate your valuable feedback and will revise the paper based on the above.

---

> > ### Comment · Reviewer_2m1M · 2024-08-14
> >
> > Thanks for the rebuttal. I have read through it and appreciate the response. I will keep my score unchanged.

---

> > > ### Author Response · Authors · 2024-08-14
> > >
> > > Thanks for your response, and we appreciate your recognition of our work.

---

### Author Rebuttal · Authors · 2024-08-07

We sincerely appreciate the valuable feedback provided by all the reviewers. We have carefully addressed their questions and concerns in our response, aiming to provide satisfactory answers.

Here we uploaded a file named "global.pdf" to show some necessary results and comparisons, which contains three tables and one figure.

In all subsequent responses to reviewers, this file is termed ***global.pdf*** for easy presentation and convenient discussion.

---

### Decision · Program_Chairs · 2024-09-25

**Decision:**

Accept (poster)

**Comment:**

The paper presents a meta multigraph-assisted disentangled graph learning framework for cognitive diagnosis (CD) in educational contexts, named DisenGCD. This new method constructs three distinct types of representations on separate disentangled graphs: student-exercise-concept interactions, exercise-concept relationships, and concept dependencies. Employing GAT-based modules, it effectively learns nuanced exercise and concept representations from these graphs. A specialized diagnostic function integrates these representations for predictive assessment. DisenGCD demonstrates superior performance over other state-of-the-art CD methods, excelling in AUC, accuracy, and RMSE metrics.

All four reviewers agree that the paper meets the acceptance criteria and have offered several constructive suggestions, including comparisons with more recent CD methods and conducting an ablation study to evaluate the impact of model parameters and the choice of graph neural network. The authors have addressed these points to the reviewers' satisfaction and are encouraged to include these additional results in the final version of the paper.